# *You Only Prune Once*: Designing Calibration-Free Model Compression With Policy Learning

**Ayan Sengupta**[*]**, Siddhant Chaudhary**[*] **& Tanmoy Chakraborty**
Department of Electrical Engineering
Indian Institute of Technology Delhi, India
`ayan.sengupta@ee.iitd.ac.in, urssidd@gmail.com, tanchak@iitd.ac.in`

## Abstract

The ever-increasing size of large language models (LLMs) presents significant challenges for deployment due to their heavy computational and memory requirements. Current model pruning techniques attempt to alleviate these issues by relying heavily on external calibration datasets to determine which parameters to prune or compress, thus limiting their flexibility and scalability across different compression ratios. Moreover, these methods often cause severe performance degradation, particularly in downstream tasks, when subjected to higher compression rates. In this paper, we propose **PruneNet**, a novel model compression method that addresses these limitations by reformulating model pruning as a policy learning process. `PruneNet` decouples the pruning process from the model architecture, eliminating the need for calibration datasets. It learns a stochastic pruning policy to assess parameter importance solely based on intrinsic model properties while preserving the spectral structure to minimize information loss. `PruneNet` can compress the LLaMA-2-7B model in just 15 minutes, achieving over 80% retention of its zero-shot performance with a 30% compression ratio, outperforming existing methods that retain only 75% performance. Furthermore, on complex multitask language understanding tasks, `PruneNet` demonstrates its robustness by preserving up to 80% performance of the original model, proving itself a superior alternative to conventional structured compression techniques. [1]

## 1 Introduction

Pre-trained Large Language models (LLMs) (Touvron et al., 2023; Zhang et al., 2022; Scao et al., 2023; OpenAI et al., 2024) have demonstrated exceptional abilities in natural language understanding and generation, creating numerous avenues of applications across a wide range of domains such as healthcare, education and finance. These deep neural models are predominantly based on the Transformer architecture (Vaswani et al., 2023) and often contain several billions of parameters. Models like GPT-4, and larger variants of ($> 65B$ parameters) LLaMA-2, and OPT can occupy as much as 350GB of memory in FP16 format, making them impractical for deploying to resource-constrained environments such as mobile or edge devices. The need for dedicated GPUs, even for inference, restricts their applicability in real-time, low-resource scenarios, creating a barrier for broader adoption in industries where speed and efficiency are crucial.

**Model compression** is a class of techniques that are widely used to reduce the computational overhead of LLMs. Two major subclasses of model compression are *quantization* and *model pruning*. While quantization is primarily used to reduce the precision points of the saved model weights, thereby reducing the memory footprint of the model, model pruning aims at pruning or dropping different neural components to make models smaller and faster during inference. Notable methods like post-training model pruning methods (Ashkboos et al., 2024; Yang et al., 2024) usually work by either removing entire components like neurons, attention heads or layers based on specific rules (*structured pruning*) or removing individual parameters resulting in an irregular sparse structure

---

[*]These authors contributed equally to this work.

[1]The source code of `PruneNet` is made public at `https://github.com/LCS2-IIITD/PruneNet`.

| Method | Sparsity | Effective Sparsity | FLOPs | Avg. Zero-shot Acc |
|---|---|---|---|---|
| Dense | 0% | 0.0% | 1.35e+13 (1.00x) | 69.0 |
| SliceGPT | 20% | 9.4% | 1.23e+13 (1.10x) | 58.2 |
| PruneNet | | **12.0%** | **1.18e+13 (1.15x)** | **61.7** |
| SliceGPT | 25% | 15.3% | 1.14e+13 (1.18x) | 55.5 |
| PruneNet | | **16.0%** | **1.13e+13 (1.20x)** | **58.6** |
| SliceGPT | 30% | **21.4%** | **1.07e+13 (1.27x)** | 51.5 |
| PruneNet | | 19.0 % | 1.09e+13 (1.24x) | **55.5** |

Table 1: **A summary of the experimental results.** We highlight the effective sparsity ratio, along with total FLOPs (Floating Point Operations) and average zero-shot accuracy for different sparsity ratios with the LLaMA-2-7B model (see Table 2 for more details). *Effective sparsity* can be calculated as the ratio of the total number of parameters in the compressed model and that of the uncompressed model. SliceGPT achieves less effective sparsity with high FLOPs at a lower sparsity ratio ($< 25\%$) as it learns the pruned parameters with a learnable network and retains them within the LLM. On the other hand, PruneNet decouples the compression process from the LLM, thereby achieving higher effective sparsity with lower FLOPs. Post-compression performance drop from the *dense* (uncompressed or compression ratio $0\%$) LLaMA-2-7B model is also significantly higher for SliceGPT (average drop of $13.9\%$, as compared to a drop of $10.7\%$ of PruneNet).

(*unstructured pruning*). For instance, SliceGPT (Ashkboos et al., 2024) uses orthogonal transformations on pre-trained model parameters to slice off a contiguous block of rows and columns, reducing the overall size of the models. However, these methods face notable limitations, particularly their reliance on calibration data and hardware-specific optimizations, constraining their flexibility and scalability. Most notably, the heavy dependence on calibration data limits the applicability of existing pruning methods. Over-reliance on calibration datasets makes the model compression methods vulnerable to data quality issues and reduces their trustworthiness for performance-sensitive applications like finance or healthcare. Furthermore, the need to repeatedly calibrate models for different compression ratios hinders their efficiency, making it difficult to adapt to varying computational resource constraints.

**Our Contribution.** To address these challenges, we introduce **PruneNet**, a novel structured pruning technique that eliminates the need for calibration data and enhances flexibility across varying compression ratios. Unlike traditional methods, PruneNet reformulates model pruning as a *policy learning process*, leveraging a reusable policy learner that decouples parameter importance assessment from the model architecture itself. This allows for rapid pruning without repeated retraining, enabling the same pruning strategy to be applied at multiple compression ratios. The method also minimizes the information loss after model compression by preserving the uncompressed and compressed models' spectrum structure. Our empirical analysis demonstrates that PruneNet can compress the LLaMA-2-7B model in just 15 minutes — 50% faster than existing methods like SliceGPT while retaining up to 95% of the original model's performance across various tasks. Remarkably, even without recovery fine-tuning, PruneNet preserves 84% of the zero-shot accuracy on commonsense reasoning tasks (c.f. Table 1), outperforming SliceGPT by a significant margin. Furthermore, even smaller LLMs such as OPT-125M exhibit post-compression performance stability, suggesting that PruneNet is not only efficient but also robust across models of varying sizes.

## 2 RELATED WORK

Model pruning is a widely-used technique to reduce the number of parameters in a model, enhancing both speed and efficiency. It can be broadly categorized into two classes – unstructured and structured pruning. Unstructured pruning removes individual weights, as seen in SparseGPT (Frantar & Alistarh, 2023), which leverages Hessian matrix inversion to identify and eliminate less critical weights. However, unstructured pruning often requires hardware-specific optimizations and may not always result in significant computational gains (Yang et al., 2024; Wang et al., 2024b). In contrast, structured pruning removes entire channels or components, making it more compatible with various hardware setups. For example, LLM-Pruner (Ma et al., 2023b) evaluates weight group importance and uses LoRA fine-tuning to recover lost accuracy. While structured pruning is more hardware-friendly, it can lead to greater accuracy loss at higher compression ratios. Methods like Layer Collapse (Yang et al., 2024) take a layer-wise approach, merging parameters of adjacent lay-

ers to achieve up to 50% compression without extensive retraining. Recent advancements in post-training compression methods, such as SliceGPT (Ashkboos et al., 2024) and SVD-LLM (Wang et al., 2024b), aim to maintain performance while reducing model size. SliceGPT is a structured pruning method that compresses LLMs by *slicing off* entire rows and columns of weight matrices, using orthogonal transformations to reduce the embedding dimensions. SVD-LLM, on the other hand, applies singular value decomposition with layer-wise updates, ensuring minimal accuracy loss even under high compression, outperforming previous methods like ASVD (Yuan et al., 2023) and FWSVD (Hsu et al., 2022).

Majority of these structured pruning methods rely heavily on external calibration datasets, making them sensitive to data quality. For instance, the LLaMA-2-7B model, compressed with SliceGPT, drops zero-shot performance by 6% when calibrated with the WikiText2 dataset instead of the Alpaca dataset. In contrast, our proposed `PruneNet` method does not rely on any calibration data and captures the parameter importance as a learnable policy. The policy learner model, being independent of the original model, offers great flexibility and can be reused to compress different models at different compression ratios.

## 3 BACKGROUND

### 3.1 TRANSFORMER ARCHITECTURE

The Transformer architecture (Vaswani et al., 2023) has been widely adopted to achieve state-of-the-art results in a wide range of natural language tasks, including natural language understanding and generative language modeling. Each layer of a Transformer model includes two core components: the multi-headed self-attention layer followed by a feed-forward layer (FFN), which are separated by LayerNorm (Ba et al., 2016) blocks and residual connections (He et al., 2015). In most Transformer architectures, the FFN module is a two-layered multi-layer perceptron (MLP), which can be mathematically represented as follows:

$$\text{FFN}(\boldsymbol{X}) = \sigma(\boldsymbol{X}\boldsymbol{W}_{\text{up}}^T + \boldsymbol{b}_{\text{up}})\boldsymbol{W}_{\text{down}}^T + \boldsymbol{b}_{\text{down}} \tag{1}$$

where $\sigma$ is a non-linear activation function, $\boldsymbol{X} \in \mathbb{R}^{B \times N \times d_{\text{hidden}}}$ is a batch of inputs with $B$ being the batch size, $N$ denotes the sequence length, $d_{\text{hidden}}$ is the hidden dimension of the architecture, $\boldsymbol{W}_{\text{up}} \in \mathbb{R}^{d_{\text{intermediate}} \times d_{\text{hidden}}}$ is the up-projection matrix (*i.e.,* $d_{\text{intermediate}} > d_{\text{hidden}}$), $\boldsymbol{W}_{\text{down}} \in \mathbb{R}^{d_{\text{hidden}} \times d_{\text{intermediate}}}$ is the down-projection matrix, and $\boldsymbol{b}_{\text{up}} \in \mathbb{R}^{d_{\text{intermediate}}}$ and $\boldsymbol{b}_{\text{down}} \in \mathbb{R}^{d_{\text{output}}}$ are the corresponding biases. The first and second layers of MLP are also referred to as $\text{FFN}_1$ and $\text{FFN}_2$, respectively; we point out that all these notations just refer to the two MLP layers, and for our purposes, can be used interchangeably. While most LLMs such as OPT (Zhang et al., 2022), Falcon (Almazrouei et al., 2023) and Phi (Gunasekar et al., 2023) have two weight matrices in the FFN layers, models like LLaMA (Touvron et al., 2023) have three matrices in their FFN layers: the up-projection matrix ($\boldsymbol{W}_{\text{up}}$), the down-projection matrix ($\boldsymbol{W}_{\text{down}}$) and a gated-projection matrix $\boldsymbol{W}_{\text{gate}}$.

### 3.2 INTRINSIC MODEL COMPRESSION

Several existing structured pruning methods (*e.g.,* SliceGPT) prune matrices by *intrinsically computing parameter importance*. Specifically, for every attention and FFN layer, SliceGPT adds an additional matrix of dimension $d_{\text{hidden}} \times (1 - r) \cdot d_{\text{hidden}}$ and $d_{\text{hidden}} \times (1 - r) \cdot d_{\text{intermediate}}$, for a compression ratio of $r$. It turns out that for small compression ratios, such techniques end up adding more parameters to the model than they remove, leading to an overall increase in the total parameter count of the model. We formalize this fact in the following lemma.

**Lemma 3.1** (**Limitations of Intrinsic Model Compression**). *Given an LLM with hidden dimension $d_{hidden}$ and intermediate FFN dimension $d_{intermediate}$, any intrinsic model compression method that introduces new parameters within the model will reduce model size only if the compression ratio $r > \frac{d_{hidden} + d_{intermediate}}{5d_{hidden} + 3d_{intermediate}}$.* [2]

For Phi-2, $d_{\text{hidden}} = 2,560$ and $d_{\text{intermediate}} = 10,240$. Therefore, for any compression ratio $r < 29.4\%$, the compressed model can have negative effective compression, defying the purpose of model compression.

---

[2] See Appendix C.1 for the proof.

## 3.3 Effects of Slicing on the Spectrum of a Matrix

In a slicing operation (popularly used in methods like SliceGPT), we drop entire row(s) or column(s) from the weight matrices of a model. We highlight two key results with slicing on the spectrum of a weight matrix.

For any matrix $W \in \mathbb{R}^{n \times m}$, the matrix $W^T W$ (also known as the *Gram Matrix* of $W$) is a positive-semidefinite matrix, thus admitting only real, non-negative eigenvalues. Moreover, the non-zero singular values of $W$ are precisely square roots of the non-zero eigenvalues of $W^T W$. Throughout the paper, we will refer to the set of eigenvalues of $W^T W$ as the *spectrum* of $W$.

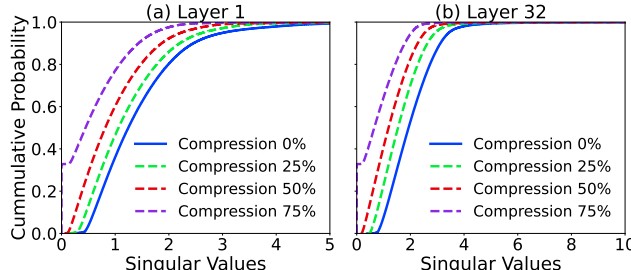

Figure 1: Empirical cumulative distribution function (ECDF) of singular values of $FFN_1$ layer weights at layers 1 (a) and 32 (b) of the LLaMA-2-7B model. The right skewness of the distribution at higher sparsity ratio highlights the diminishing property of singular values at reduced dimensions.

**Theorem 3.2** (**Poincaré Separation Theorem**). *Let $A \in \mathbb{R}^{n \times n}$ be symmetric matrix, and let $B \in \mathbb{R}^{n \times r}$ be a semi-orthogonal matrix, i.e., $B^T B = I_r$, where $1 \leq r \leq n$ and $I_r \in \mathbb{R}^{r \times r}$ is the identity matrix. Let $\lambda_1 \leq \lambda_2 \leq \cdots \leq \lambda_n$ be the eigenvalues of $A$ and $\mu_1 \leq \mu_2 \leq \cdots \leq \mu_r$ be the eigenvalues of $B^T A B$. Then*

$$\lambda_i \leq \mu_i \leq \lambda_{n-r+i}$$

*for $1 \leq i \leq r$. In particular,*

$$\min_{1 \leq i \leq n} \lambda_i \leq \min_{1 \leq j \leq r} \mu_j \leq \max_{1 \leq j \leq r} \mu_j \leq \max_{1 \leq i \leq n} \lambda_i \tag{2}$$

*i.e., the range of eigenvalues of $B^T A B$ is a subset of the range of eigenvalues of $A$.* [3]

**Corollary 3.3** (**Slicing shrinks the range of the spectrum**). *Let $W \in \mathbb{R}^{n \times d}$ be a weight matrix, and let $W' \in \mathbb{R}^{m \times d}$ be a matrix obtained by slicing off rows of $W$ so that $m \leq n$. Then, the range of singular values of $W'$ is a subset of the range of singular values of $W$.* [4]

Corollary 3.3 suggests that if we remove any set of rows from any weight matrix in an LLM, the distribution of singular values of the matrix becomes more right-skewed. To understand this phenomenon, we illustrate the spectrum of $FFN_1$ from layer 1 and 32 of the LLaMA-2-7B model in Figure 1. This observation encourages us to learn a compression model to minimize the distributional shift after compression to retain the performance of the uncompressed model.

## 4 Methodology

In this section, we elaborate on **PruneNet**, our proposed policy-driven **prun**ing-obj**e**cted **net**work. Corollary 3.3 lays the foundation of our method, which aims at minimizing the spectral distributional shift between post and pre-compression LLMs, thereby achieving the purpose of model pruning and maintaining the model's superiority. Figure 2 illustrates the overall PruneNet framework.

Although PruneNet is flexible and can be used to compress any neural module, we focus only on the FFN layers of LLMs as these layers contribute to the most parameters in an LLM with the highest density structure. For instance, in LLaMA-2-7B, FFN layers collectively contribute to $64\%$ of the parameters, whereas the self-attention layers contribute to $32\%$ of the parameters. Moreover, due to the multi-headed structure, the density (number of neurons directly connected to a parameter) in self-attention is only $27\%$, whereas it is $100\%$ in FFN layers. Additionally, only FFN layers are responsible for non-linear activation. Mirzadeh et al. (2023) highlighted that non-linear activations comprise a significant computational cost within an LLM. Therefore, we make a logical assumption only to compress the FFN layers of an LLM to achieve the highest effective sparsity.

---

[3]See Theorem 11.10 of Magnus (2019) for the proof.
[4]See Appendix C.2 for the proof.

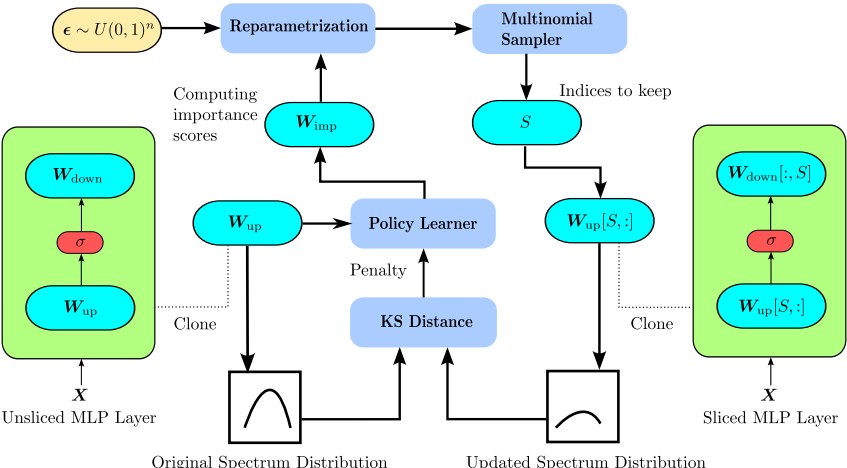

Figure 2: **A schematic diagram of `PruneNet`**. A policy learner is used to learn the row indices of an $\text{FFN}_1$ weight matrix to prune. The policy learner is trained with policy gradient with penalty calculated with the Kolmogorov-Smirnov (KS) distance between the uncompressed and compressed matrix singular value distributions. $\text{FFN}_2$ is pruned by columns with the same indices learned by the policy learner. Biases, $\boldsymbol{b}_{\text{up}}$ and $\boldsymbol{b}_{\text{down}}$, are excluded from the diagram, but are also pruned.

## 4.1 IDENTIFYING PARAMETERS TO COMPRESS

The foundational block of `PruneNet` is the policy learner model. The policy learner is a simple MLP, which is used to determine the row indices of an $\text{FFN}_1$ weight matrix, $\boldsymbol{W}_{\text{up}}$ to prune (see Section 3.1 for notation). We refer to this process as *selective pruning* – it does not assume the continuity of columns or rows in the selection. The policy learner is used to prune the $\text{FFN}_1$ weight matrices of *all* MLP layers in the model.

Consider an $\text{FFN}_1$ weight matrix $\boldsymbol{W}_{\text{up}} \in \mathbb{R}^{n \times d}$ of the model. To prune rows of this matrix, the policy learner aims to compute a vector $\boldsymbol{W}_{\text{imp}} \in \mathbb{R}^n$ of *importance scores* for each row; all importance scores are in $(0, 1)$. The policy learner model has two trainable parameters: an intermediate projection matrix $\boldsymbol{W}_{\text{inter}} \in \mathbb{R}^{n \times d}$ and a final projection matrix $\boldsymbol{W}_{\text{proj}} \in \mathbb{R}^{1 \times n}$. The computation of $\boldsymbol{W}_{\text{imp}}$ can be described as follows:

$$\boldsymbol{W}' = \boldsymbol{W}_{\text{up}} \boldsymbol{W}_{\text{inter}}^T$$
$$\boldsymbol{W}_{\text{imp}} = \sigma(\boldsymbol{W}_{\text{proj}} \boldsymbol{W}') \tag{3}$$

Note that $\boldsymbol{W}' \in \mathbb{R}^{n \times n}$ can capture the interaction between *all rows* of the matrix to assess the interdependence among them. We do not assume any bias while computing importance scores; hence, we do not use any bias parameter in the policy learner model.

Using the importance scores $\boldsymbol{W}_{\text{imp}}$, we determine the rows of $\boldsymbol{W}_{\text{up}}$ to prune. Given a compression ratio $r \in [0, 1]$, we sample a set $S$ containing $(1 - r) \cdot n$ indices from the set $\{1, 2, \ldots, n\}$ from a *multinomial distribution* with event importance[5] $\tilde{\boldsymbol{W}}_{\text{imp}}$, calculated as follows:

$$\tilde{\boldsymbol{W}}_{\text{imp}} = \sigma(\log \boldsymbol{\epsilon} - \log (1 - \boldsymbol{\epsilon}) + \log (\boldsymbol{W}_{\text{imp}}) - \log (1 - \boldsymbol{W}_{\text{imp}})) \tag{4}$$

where above, $\boldsymbol{\epsilon} \in \mathbb{R}^n$ is a random vector, each of whose entries are sampled from the uniform distribution, $U(0, 1)$. Equation (4) uses the well-known *reparametrization trick* (Kingma, 2013; Maddison et al., 2016) to ensure that the sampling process is differentiable w.r.t the importance scores $\boldsymbol{W}_{\text{imp}}$. [6]

To get the pruned matrix, we prune all indices not in $S$:

$$\boldsymbol{W}_{\text{up}}^{\text{compressed}} = \boldsymbol{W}_{\text{up}}[S, :] \tag{5}$$

---

[5]Event probabilities can be computed by normalizing event importances.

[6]The formulation in Equation 4 can be used to simulate a Bernoulli random variable with mean given by $\boldsymbol{W}_{\text{imp}_i}$, for $i \in [n]$. See Appendix C.3 for the proof.

While compressing an LLM, we only perform the sampling for the $FFN_1$ weight matrix. The same set of sampled indices is reused for the $FFN_2$ matrix by using $[:, S]$ to ensure that the dropped dimensions match between the two matrices. Similarly, the bias terms, $\boldsymbol{b}_{\text{up}}$ and $\boldsymbol{b}_{\text{down}}$, are pruned accordingly.

## 4.2 PENALTY COMPUTATION FOR THE POLICY LEARNER

We calculate the singular values of the original weight matrix $\boldsymbol{W}_{\text{up}}$ as $\{\lambda_i\}_{i=1}^n$ with $\lambda_1 \geq \lambda_2 \cdots \geq \lambda_n$. Similarly, for the compressed matrix $\boldsymbol{W}_{\text{up}}^{\text{compressed}}$, we obtain the singular values $\{\mu_i\}_{i=1}^{\bar{n}}$, where $\bar{n} = (1 - r) \cdot n$. To calculate the distributional difference between the spectrums, we calculate the Kolmogorov-Smirnov (KS) distance, defined as:

$$\mathcal{D} = \sup_x |F_{1,n}(x) - F_{2,\bar{n}}(x)| \tag{6}$$

where $F_{1,n}$ and $F_{2,\bar{n}}$ are the empirical distributions of $\{\lambda_i\}_{i=1}^n$ and $\{\mu_i\}_{i=1}^{\bar{n}}$, respectively. The KS distance computes the supremum of the distance between the two distributions at any point in their support. The policy learner uses the distance measure as the penalty to minimize.

## 4.3 POLICY OPTIMIZATION

The final step in `PruneNet` is to optimize the policy learner to minimize the expected penalty defined in Equation 6. Given an $L$-layered LLM, we consider each $FFN_1$ matrix weight $\boldsymbol{W}_i$ as states and the selected indices $S_i$ as the action obtained by the policy learner. Therefore, we can construct a trajectory $\boldsymbol{W}_1, S_1, \boldsymbol{W}_2, S_2, \ldots, \boldsymbol{W}_L, S_L$. Figure 1 suggests that typically, in an LLM, the later layers have higher singular values. Therefore, the penalty could be higher for the later layers. Moreover, the later layers of LLMs contain the most semantic information (Yang et al., 2024), needing them to preserve the most. Thus, for each LLM layer $l$, we calculate the discounted future penalty as:

$$G_l = \sum_{k=0}^{\infty} \gamma^k \mathcal{D}_{l+k+1}. \tag{7}$$

For a given state $\boldsymbol{W}_l$ ($FFN_1$ weight at layer $l$), we use the policy learner to calculate action probability using Equation 4, sample the action $S_l$ (set of rows to select) and obtain the compressed matrix $\boldsymbol{W}_l^{\text{compressed}}$. Using Equations 6 and 7 we calculate the penalty. Finally, using the REINFORCE (Williams, 1992) algorithm, we compute the gradient and learn the policy model.

# 5 EXPERIMENTAL RESULTS

## 5.1 EXPERIMENTAL SETUP

We use `PruneNet` for compressing different LLMs, including — LLaMA-2-7B (Touvron et al., 2023), LLaMA-1-7B, Phi-2 (Gunasekar et al., 2023) and OPT (Zhang et al., 2022) (125M, 2.7B and 6.7B variants). For the zero-shot performance evaluation, we use five commonsense reasoning tasks — PIQA (Bisk et al., 2020), WinoGrande (Sakaguchi et al., 2021), HellaSwag (Zellers et al., 2019), ARC-e and ARC-c (Clark et al., 2018), using the LM Evaluation Harness suite (Gao et al., 2024) and the MMLU benchmark (Hendrycks et al., 2020) [7]. For the policy learner model, we consider the discount factor, $\gamma = 0.99$ and use the AdamW (Loshchilov, 2017) optimizer with a learning rate of $5e^{-4}$ and a maximum of 20 episodes. The total number of trainable parameters in the policy learner is 45M ($\sim 0.67\%$) for the LLaMA-2-7B model. We evaluate the compression performance of `PruneNet` primarily against the state-of-the-art baseline SliceGPT (Ashkboos et al., 2024). Additionally, we also consider other competitive structured pruning methods — LaCo (Yang et al., 2024), SVD-LLM (Wang et al., 2024b), ASVD (Yuan et al., 2023), LLM Pruner (Ma et al., 2023b) and ShortGPT (Men et al., 2024) for our evaluation.[8] All the experiments were performed on a single Nvidia A100-40GB GPU.

---

[7]All the task descriptions are provided in Section D of the Appendix.

[8]Due to unavailability of LaCo and ShortGPT baseline source codes, we were able to compare these methods only on selected tasks.

| Model | Comp. Ratio | Method | PIQA | WinoGrande | HellaSwag | ARC-e | ARC-c | Avg. |
|---|---|---|---|---|---|---|---|---|
| LLaMA-2-7B | 0% | Dense | 79.11 (100%) | 69.06 (100%) | 75.99 (100%) | 74.58 (100%) | 46.25 (100%) | 69.00 (100%) |
| | 20% | SliceGPT | 69.42 (88%) | 65.11 (94%) | 59.04 (78%) | 59.76 (80%) | **37.54 (81%)** | 58.17 (84%) |
| | | PruneNet | **75.30 (95%)** | **65.51 (95%)** | **66.43 (87%)** | **63.80 (85%)** | 37.29 (81%) | **61.67 (89%)** |
| | 25% | SliceGPT | 66.87 (84%) | **63.38 (92%)** | 54.16 (71%) | 58.46 (78%) | 34.56 (75%) | 55.48 (80%) |
| | | PruneNet | **72.09 (91%)** | 62.43 (90%) | **62.33 (82%)** | **60.14 (81%)** | **36.18 (78%)** | **58.63 (85%)** |
| | 30% | SliceGPT | 63.55 (80%) | **61.33 (89%)** | 49.62 (65%) | 51.77 (69%) | 31.23 (67%) | 51.50 (75%) |
| | | PruneNet | **71.11 (90%)** | 61.09 (88%) | **58.30 (77%)** | **53.20 (71%)** | **33.53 (72%)** | **55.45 (80%)** |
| Phi-2 | 0% | Dense | 79.11 (100%) | 75.77 (100%) | 73.83 (100%) | 78.32 (100%) | 54.18 (100%) | 72.24 (100%) |
| | 20% | SliceGPT | 71.87 (91%) | 67.80 (89%) | 57.76 (78%) | 58.00 (74%) | 35.32 (65%) | 58.15 (80%) |
| | | PruneNet | **74.37 (94%)** | **70.80 (93%)** | **65.53 (89%)** | **74.71 (95%)** | **47.53 (88%)** | **66.59 (92%)** |
| | 25% | SliceGPT | 69.21 (88%) | 65.35 (86%) | 52.40 (71%) | 53.7 (69%) | 31.66 (58%) | 54.46 (75%) |
| | | PruneNet | **74.37 (94%)** | **68.98 (91%)** | **62.18 (84%)** | **70.54 (90%)** | **44.45 (82%)** | **64.10 (89%)** |
| | 30% | SliceGPT | 65.94 (83%) | 63.14 (83%) | 47.56 (64%) | 53.03 (68%) | 30.29 (56%) | 51.99 (72%) |
| | | PruneNet | **72.80 (92%)** | **67.48 (89%)** | **56.80 (77%)** | **67.55 (86%)** | **40.61 (75%)** | **61.05 (84%)** |

Table 2: Comparison of SliceGPT and `PruneNet` on generative tasks without recovery fine-tuning. We report the performance recovery from the uncompressed model in parenthesis. Tables 21 and 22 in Appendix E.3 highlight the zero-shot performance for the other LLMs compressed with `PruneNet` and SliceGPT.

| Model | Comp. Ratio | RFT | PIQA | WinoGrande | HellaSwag | ARC-e | ARC-c | Avg. |
|---|---|---|---|---|---|---|---|---|
| LLaMA-2-7B | 20% | ✗ | 75.30 | 65.51 | 66.43 | 62.25 | 36.26 | 61.15 |
| | | ✓ | 74.76 | 66.22 | 68.37 | 63.93 | 38.40 | **62.34** |
| | 25% | ✗ | 72.09 | 62.43 | 62.33 | 60.14 | 36.18 | 58.63 |
| | | ✓ | 74.37 | 63.77 | 65.71 | 60.65 | 35.75 | **60.05** |
| | 30% | ✗ | 71.11 | 61.09 | 58.30 | 53.20 | 32.94 | 55.33 |
| | | ✓ | 72.20 | 62.90 | 63.21 | 53.37 | 33.70 | **57.08** |
| Phi-2 | 20% | ✗ | 74.37 | 70.80 | 65.53 | 74.71 | 47.53 | **66.59** |
| | | ✓ | 76.17 | 71.51 | 63.28 | 70.50 | 46.42 | 65.58 |
| | 25% | ✗ | 74.37 | 68.98 | 62.18 | 70.54 | 44.45 | 64.10 |
| | | ✓ | 73.39 | 69.22 | 61.53 | 71.38 | 45.73 | **64.25** |
| | 30% | ✗ | 72.80 | 67.48 | 56.80 | 67.55 | 40.61 | **61.05** |
| | | ✓ | 71.49 | 63.93 | 58.18 | 61.78 | 37.80 | 58.34 |

Table 3: Performance of LLaMA-2-7B and Phi-2 under different compression ratios without (marked as ✗) and with (marked as ✓) recovery fine-tuning (RFT) on the WikiText2 dataset. Results without RFT are the same as the ones reported in Table 2.

## 5.2 RESULTS

Table 2 reports[9] the zero-shot performance of LLaMA-2-7B and Phi-2 models after being compressed with `PruneNet` and SliceGPT (the best baseline) at different compression ratios. The average performance of the uncompressed LLaMA model is $69\%$, out of which $85\%$ is preserved after compression with `PruneNet` with a maximum drop by $20\%$. On the other hand, at $30\%$ compression ratio, SliceGPT drops the performance by $25\%$. Similarly, for the Phi-2 model, the maximum performance drop is $16\%$ with `PruneNet`, which is significantly higher ($28\%$) for SliceGPT. We report the zero-shot performance of other compressed LLMs, i.e., OPT-125M, OPT-2.7B and OPT-6.7B in Table 21 of Appendix E.3. A one-sided Kolmogorov-Smirnov (KS) test suggests ($p$-value $< 0.05$) that the performance drop exhibited by `PruneNet` is significantly lower than that of SliceGPT across all the LLMs. Interestingly, for different LLMs, the performance remains stable across different compression ratios (standard deviation of $2.03$), whereas, for SliceGPT, the performance drops significantly with a higher compression rate (standard deviation of $2.54$). Even on complex multitask language understanding tasks (c.f. Table 23 in Appendix E.3), LLMs compressed with `PruneNet` exhibit high quality stability at different compression ratios. Interestingly, on tasks like formal logic and global facts, compressed LLaMA models with `PruneNet` even outperform the uncompressed ones.

Recovery fine-tuning (RFT) is a common trick to regain performance drop after compression. To understand the importance of RFT on the effectiveness of `PruneNet`, we report the zero-shot performance of compressed LLaMA and Phi-2 models after fine-tuning on the WikiText2 (Merity et al., 2016) dataset in Table 3. For fine-tuning, we use LoRA adapters (Hu et al., 2022) with rank 8. Interestingly, RFT has only a marginal impact of $1.5\%$ on the compressed LLaMA model, which highlights the robustness of our method. Remarkably, the importance of RFT remains the same for a higher compression rate. On the other hand, with Phi-2, the performance drops after RFT in several cases. This result validates the robustness of `PruneNet` but also an appreciation of the pre-

---

[9]Further ablation studies and a detailed experimental analysis can be found in section E of the appendix.

| Comp. Ratio | Method | PIQA | Hellaswag | Average |
|---|---|---|---|---|
| 0% | Dense | 77.91 | 71.26 | 74.58 |
| | PruneNet | 71.11 | **58.30** | **64.70** |
| | LLM Pruner (Ma et al., 2023a)† | **71.22** | 56.46 | 63.84 |
| 30% | SliceGPT (Ashkboos et al., 2024)† | 66.21 | 50.27 | 58.24 |
| | LaCo (Yang et al., 2024)† | 69.80 | 55.69 | 62.74 |
| | ShortGPT (Men et al., 2024)† | 66.43 | 53.02 | 59.72 |

Table 4: Comparison of different model compression methods on generative tasks with the LLaMA-2-7B model without recovery fine-tuning. Results marked with '†' are taken from Men et al. (2024).

| Comp. Ratio | Method | PIQA | HellaSwag | WinoGrande | ARC-e | ARC-c | Average |
|---|---|---|---|---|---|---|---|
| | PruneNet | **76** | 65 | **62** | **64** | 36 | **61.0** |
| | SliceGPT (Ashkboos et al., 2024) | 72 | 61 | **62** | **64** | 35 | 59.0 |
| 20% | LLM-Pruner (Ma et al., 2023b) | **76** | 67 | 61 | 61 | **37** | 60.4 |
| | ASVD (Yuan et al., 2023) | 68 | 41 | 64 | 53 | 27 | 50.6 |
| | SVD-LLM (Wang et al., 2024a) | 69 | 43 | 63 | 58 | 29 | 52.4 |
| | PruneNet | **70** | **57** | 59 | **52** | 31 | **53.8** |
| | SliceGPT (Ashkboos et al., 2024) | 69 | 56 | **61** | 49 | **32** | 53.3 |
| 30% | LLM-Pruner (Ma et al., 2023b) | 69 | **57** | 59 | 48 | 30 | 52.6 |
| | ASVD (Yuan et al., 2023) | 65 | 37 | 53 | 43 | 25 | 44.6 |
| | SVD-LLM (Wang et al., 2024a) | 65 | 37 | 59 | 48 | 26 | 47.0 |

Table 5: Comparison of different model compression techniques with the LLaMA-1-7B model without recovery fine-tuning.

training objective of the small language models such as Phi-2 that uses specialized curated datasets for pre-training.

Table 4 reports the performance of the LLaMA-2-7B model compressed with different structured pruning methods. At the same compression ratio, LLaMA-2-7B compressed with PruneNet achieves 1.6% higher performance than the other methods like LLM-Pruner, SliceGPT and LaCo. Our comparison against other contemporary structured pruning methods like SVD-LLM and ASVD on a different model – LLaMA-1-7B (c.f. Table 5) also highlights a similar trend. ASVD and SVD-LLM work on a similar spectral preservation objective while compressing LLMs. However, it is worth noting that these methods preserve the information loss on the output rather than the compressed model, therefore needing access to calibration datasets. Even after calibration, these methods significantly underperform PruneNet (average difference of 8.5%). Typically, SliceGPT and LLM-Pruner perform better than these SVD-based model compression techniques, with SliceGPT being most robust baseline. However, it is worth noting that, our method surpasses all these baselines with consistent margin across different compression ratios.

**Can we reuse the policy learner?** We evaluate if an LLM trained with the policy learner at a higher compression ratio can perform well in zero-shot commonsense tasks at lower compression ratios. Precisely, we assess the flexibility of PruneNet at different compression ratios. Table 6a highlights the zero-shot performance of the compressed LLaMA model, reusing the policy learned at a compression ratio of 40%. It turns out that PruneNet is robust in terms of the learned policy,

| Comp. Ratio | PIQA | HellaSwag | WinoGrande | ARC-e | ARC-c | Average |
|---|---|---|---|---|---|---|
| 10% | 76.88 (77.09) | 71.06 (71.06) | 65.67 (66.14) | 68.14 (68.14) | 40.87 (40.87) | 64.52 (64.66) |
| 20% | 75.30 (75.30) | 66.10 (66.43) | 65.50 (65.51) | 60.30 (63.80) | 36.30 (37.29) | 60.70 (61.67) |
| 30% | 69.21 (71.11) | 57.92 (58.30) | 61.09 (61.09) | 51.89 (53.20) | 32.94 (33.53) | 54.61 (55.45) |

(a) High compression → low compression policy transfer

| Comp. Ratio | PIQA | HellaSwag | WinoGrande | ARC-e | ARC-c | Average |
|---|---|---|---|---|---|---|
| 20% | 75.14 (75.30) | 66.4 (66.43) | 63.9 (65.51) | 63.8 (63.80) | 37.29 (37.29) | 61.31 (61.67) |
| 30% | 70.18 (71.11) | 57.81 (58.30) | 59.83 (61.09) | 53.11 (53.20) | 33.53 (33.53) | 54.89 (55.45) |
| 40% | 65.13 (66.32) | 48.59 (48.26) | 55.25 (55.80) | 41.67 (48.61) | 26.96 (28.41) | 47.52 (49.48) |

(b) Low compression → high compression policy transfer

Table 6: Zero-shot performance of the LLaMA-2-7B model compressed with the policy learned at 40% (a) and 10% (b) compression ratios. The numbers reported in parentheses are the results obtained with the model compressed with the same ratio as the policy learner.

| Model | PIQA | HellaSwag | WinoGrande | ARC-e | ARC-c | Average |
|---|---|---|---|---|---|---|
| LLaMA-2-7B | 73.18 (75.30) | 65.35 (66.43) | 65.10 (65.51) | 58.88 (63.80) | 35.07 (37.29) | 59.52 (61.67) |
| Phi-2 | 74.65 (74.37) | 65.88 (65.53) | 69.53 (70.80) | 73.70 (74.71) | 47.87 (47.53) | 66.35 (66.59) |

Table 7: Zero-shot performance of the LLaMA-2-7B and Phi-2 models when different layers compressed at different compression ratio with maximum being $40\%$. For a fair comparison, we compare these results with models compressed at a fixed compression ratio of $20\%$.

| Comp. Ratio | RFT Dataset | PIQA | WinoGrande | HellaSwag | ARC-e | ARC-c | Average |
|---|---|---|---|---|---|---|---|
| | Alpaca | 72.73 | 62.25 | 66.45 | 61.52 | 42.15 | 61.02 |
| 20% | PTB | 73.07 | 63.38 | 65.05 | 64.56 | 36.69 | 60.55 |
| | WikiText2 | 74.76 | 66.22 | 69.38 | 65.61 | 39.25 | 63.04 |
| | Alpaca | 75.79 | 62.35 | 65.48 | 60.94 | 39.16 | 60.74 |
| 25% | PTB | 73.47 | 63.28 | 63.56 | 62.79 | 36.19 | 59.86 |
| | WikiText2 | 74.37 | 66.46 | 65.71 | 60.82 | 36.60 | 60.79 |
| | Alpaca | 72.14 | 62.75 | 62.38 | 55.43 | 37.03 | 57.95 |
| 30% | PTB | 71.11 | 62.67 | 59.60 | 58.12 | 35.07 | 57.31 |
| | WikiText2 | 73.01 | 63.46 | 63.21 | 60.14 | 35.92 | 59.15 |

Table 8: Results obtained by LLaMA-2-7B after recovery finetuning on Alpaca, PTB and WikiText2 datasets with different compression ratios.

where the performance only drops marginally (maximum drop $< 1\%$ with $p$-value $< 0.05$) with transferable policy. Similar observations are found (c.f. Table 6b) when the policy learned at lower compression ratio is reused to perform compression at higher compression ratio. Remarkably, the average drop with the different policy is only $0.96\%$, with maximum margin observed only at higher compression ratios. Interestingly, even under transferred policies, the results are still $3\%$ better than SliceGPT. Therefore, if we cannot retrain the policy learner at different compression ratios under certain circumstances, reusing another policy trained for compressing the same model at various compression ratios is still feasible. This transferable property of `PruneNet` can have tremendous practical implications that most existing contemporary compression methods fail to display.

**Can we use different compression ratios in different layers?** The policy learner of `PruneNet` enables us to use different compression ratios at different LLM layers. To empirically validate its effectiveness, we evaluate LLaMA-2-7B and Phi-2 models compressed at varying compression ratios between $[0\% - 40\%]$ (average compression ratio $20\%$) at different layers. Table 7 highlights that varying compression ratios have a mixed effect on the two models. Phi-2 is highly stable, and the average accuracy drops only by $0.24\%$ with varying compression. However, for the LLaMA-7B model, the drop is more prominent, $2.15\%$.

**Does the RFT dataset matter?** Table 3 highlight that RFT has very little requirement with `PruneNet`. However, it would be interesting to understand whether the RFT datasets have any particular impact on the fine-tuned compressed model. Table 8 reports the zero-shot performance of the LLaMA-2-7B model after being fine-tuned on different datasets. It turns out that the standard deviation of performance after RFT on different datasets is only $1.3$ and can go down to even $0.52$ for certain compression ratios. Although it is assumed that instruction-tuning datasets like Alpaca are more suitable for acquiring knowledge on instruction-tuning commonsense tasks, in reality, all the datasets have a similar impact on the compressed model. This observation suggests that the robustness of `PruneNet` preserves the key information within the compressed model, therefore minimizing the need for fine-tuning on specialized datasets.

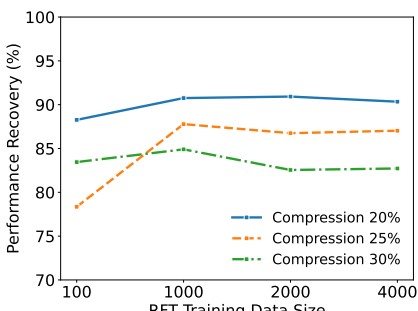

Figure 3: We highlight the performance recovery (w.r.t. the dense uncompressed model) for the LLaMA-2-7B model under different compression ratios after recovery fine-tuning with varying training data sizes.

**Does RFT data size matter?** The previous experiments suggest that RFT datasets do not significantly affect the compressed dataset's zero-shot performance. The pertinent research question is whether the RFT training data size immediately affects the compressed model's downstream perfor-

mance. To answer this, we perform RFT with varying training data sizes. Figure 3 highlights that the training data size also minimally impacts the compressed model's zero-shot downstream performance. Even with 100 training samples, a compressed LLaMA-2-7B with $< 25\%$ compression ratio can preserve up to $85\%$ of the original performance. However, 100 samples may be insufficient at a higher compression ratio of $30\%$. At higher compression rates, the compressed model quickly regains performance with more than 1000 training samples.

## 5.3 Analysis

**Does sparsity help in recovery fine-tuning?** Figure 4, shows the impact of model compression on recovery fine-tuning or any fine-tuning post-compression. Interestingly, compression can make LLMs more efficient during later fine-tuning. A highly compressed LLM usually has high training and validation loss (high perplexity) initially but can regain performance quickly even in just 50 training iterations. With a higher compression ratio, LLMs can eliminate more redundant and low-importance features, making future training and inference efficient without significant performance drops.

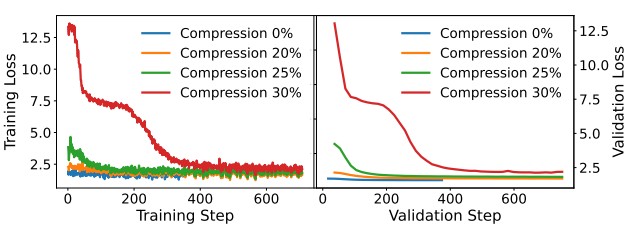

Figure 4: LLaMA-2-7B RFT training and validation loss curves for different compression ratios. We observe that sparser models achieve faster convergence and generalization.

**Computational analysis.** Table 9a exhibits the runtime complexity of `PruneNet`. To compress LLaMA-7B, `PruneNet` requires $\sim$ 15 minutes, whereas, for the same compression ratio, SliceGPT requires 29 minutes. At $30\%$ compression rate, a compressed LLaMA-7B model exhibits $73\%$ better efficiency (tokens generated per second) during inference. For a similar compression rate, SliceGPT exhibits very marginal ($7\%$) inference efficiency over the dense uncompressed model. Interestingly, for the smaller Phi-2 model, SliceGPT has poorer throughput even than the dense model, whereas, `PruneNet` manages to achieve $46\%$ higher throughput than the uncompressed model.

| Model | Params | Comp. Modules | Runtime (in sec) |
|---|---|---|---|
| LLaMA-2-7B | 6.7B | Up, Down & Gate Proj | $916 \pm 15$ |
| Phi-2 | 2.8B | FC1, FC2 | $342 \pm 8$ |

(a) Runtime of `PruneNet`

| Model | Method | Throughput (Token/sec) |
|---|---|---|
| | Dense | 11.96 |
| LLaMA-2-7B | SliceGPT | 12.82 |
| | `PruneNet` | **20.74** |
| | Dense | 20.20 |
| Phi-2 | SliceGPT | 18.48 |
| | `PruneNet` | **29.50** |

(b) Inference throughput

Table 9: (a) Time taken in seconds for compressing LLaMA-2-7B and Phi-2 models with `PruneNet` at 30% compression ratio. (b) Inference throughput (tokens generated per second) of different compression methods at $30\%$ compression ratio. The experiments were conducted on a single Nvidia A100-40GB GPU.

## 6 Conclusion

This paper introduced a novel structured pruning method, `PruneNet` for compressing LLMs. Unlike the existing compression methods, `PruneNet` learns a general policy for a given LLM and, therefore, can be reused to compress the model at any given compression rate without needing the compression model to rerun. Moreover, by utilizing the intrinsic properties of the LLM, `PruneNet` can get rid of the over-reliance on external calibration datasets and improve the stability of the compressed model on downstream tasks. While our current work explores the effectiveness of a calibration-free learnable compression policy and is applicable only for sparsifying weight matrices, evaluating its effectiveness on activation sparsity would be interesting. Even higher inference efficiency can be achieved with activation sparsity, where a similar formulation can be utilized to prune the non-linear activations with low information gain. Moreover, in the current setting, `PruneNet` is orthogonal to the concepts of quantization and, therefore, can easily be integrated with quantization methods to enhance the computational efficiency of LLMs further. In cooperation with quantization and mixed-precision training, `PruneNet` can achieve higher inference yield, with marginal performance drop, paving the way for better utilization of large language models in real-life applications.

ACKNOWLEDGMENT

T. Chakraborty acknowledges the support of the IBM-IITD AI Horizons network and Rajiv Khemani Young Faculty Chair Professorship in Artificial Intelligence.

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

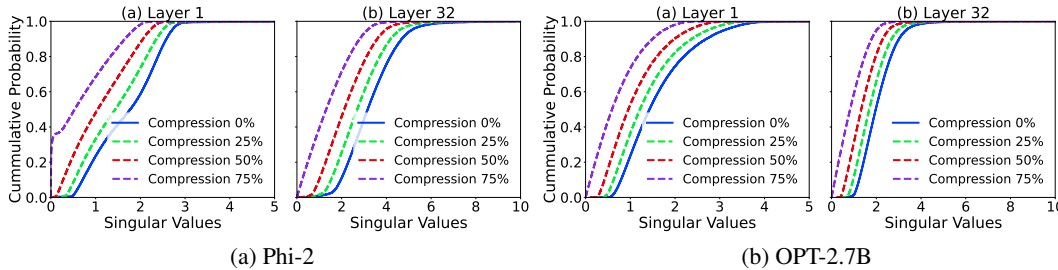

Figure 5: Spectrum of FFN1 layer for Phi-2 (a) and OPT-2.7B (b) models at different compression ratios.

Rowan Zellers, Ari Holtzman, Yonatan Bisk, Ali Farhadi, and Yejin Choi. Hellaswag: Can a machine really finish your sentence? *arXiv preprint arXiv:1905.07830*, 2019.

Susan Zhang, Stephen Roller, Naman Goyal, Mikel Artetxe, Moya Chen, Shuohui Chen, Christopher Dewan, Mona Diab, Xian Li, Xi Victoria Lin, Todor Mihaylov, Myle Ott, Sam Shleifer, Kurt Shuster, Daniel Simig, Punit Singh Koura, Anjali Sridhar, Tianlu Wang, and Luke Zettlemoyer. Opt: Open pre-trained transformer language models, 2022. URL `https://arxiv.org/abs/2205.01068`.

# A  BACKGROUND

## A.1  EFFECTS OF SLICING ON THE SPECTRUM OF A MATRIX

Weight matrices of LLMs have been known to exhibit a variation in the distribution of singular values across layers (Yuan et al., 2023). To further verify our observations from Figure 1, we conduct a similar study on the FFN1 matrices of the OPT and Phi-2 models. We plot the cumulative distribution of the singular values of these matrices and observe consistent behaviour across all three model architectures. Figure 5a and 5b highlights the spectrum of the FFN1 module at different layers for Phi-2 and OPT-2.7B models, respectively. This further strengthens our assumption of using discounted rewards in our policy learning approach.

## A.2  SPARSITY VS. EFFECTIVE SPARSITY

In structured model compression, we often choose a predetermined compression ratio to determine how much to compress a pre-trained model. We refer to this predetermined compression ratio as *sparsity ratio*. Based on the sparsity ratio, structured pruning methods prune the model components subjected to compression. Usually, the self-attention and FFN blocks of Transformer models are pruned. However, it is worth understanding that model compression does not affect all the components equally. For instance, the token embedding and final output layers of a pre-trained LLM are usually not pruned during compression. Therefore, to understand the effective impact of a model compression method on the pre-trained model as a whole, we can calculate $1 - \frac{\#\text{parameters in compressed model}}{\#\text{parameters in uncompressed model}}$, referred as *effective sparsity*. It is easy to notice that effective sparsity ≤ sparsity ratio. Lemma 3.1 highlights that for certain model compression methods and sparsity ratios, effective sparsity can even be negative, where the number of parameters in the compressed model is higher than in the uncompressed model.

# B  METHODOLOGY

We highlight the compression process with `PruneNet` in Algorithm 1.

---

**Algorithm 1** Policy-Driven Model Compression Framework (`PruneNet`)

---

**Require:** LLM with $L$ layers, FFN$_1$ weight matrices $\{\boldsymbol{W}_l\}_{l=1}^{L}$, compression ratio $r$, policy learner parameters $\boldsymbol{W}_{\text{inter}}, \boldsymbol{W}_{\text{proj}}$, discount factor $\gamma$

**Ensure:** Compressed LLM with pruned FFN layers

1: Initialize policy learner parameters
2: **for** each training step **do**
3:     **for** each layer $l = 1, 2, \ldots, L$ **do**
4:         **Compute importance scores:**
5:         $\boldsymbol{W}' \leftarrow \boldsymbol{W}_l \cdot \boldsymbol{W}_{\text{inter}}^{\top}$                        ▷ Interaction of all rows
6:         $\boldsymbol{W}_{\text{imp}} \leftarrow \sigma(\boldsymbol{W}_{\text{proj}} \cdot \boldsymbol{W}')$            ▷ Apply activation function
7:         **Sample row indices:**
8:         $\boldsymbol{\epsilon} \sim U(0,1)^n$                              ▷ Sample random vector
9:         $\tilde{\boldsymbol{W}}_{\text{imp}} \leftarrow \sigma(\log(\boldsymbol{\epsilon}) - \log(1 - \boldsymbol{\epsilon}) + \log(\boldsymbol{W}_{\text{imp}}) - \log(1 - \boldsymbol{W}_{\text{imp}}))$    ▷ Reparametrization trick
10:         Normalize $\tilde{\boldsymbol{W}}_{\text{imp}}$ to obtain probabilities
11:         $S_l \leftarrow$ Sample Multinomial$(\tilde{\boldsymbol{W}}_{\text{imp}}, (1 - r) \cdot n)$
12:         **Prune FFN$_1$:** $\boldsymbol{W}_l^{\text{compressed}} \leftarrow \boldsymbol{W}_l[S_l, :]$
13:         **Prune FFN$_2$:** $\boldsymbol{W}_{l,\text{FFN}_2}^{\text{compressed}} \leftarrow \boldsymbol{W}_{l,\text{FFN}_2}[:, S_l]$
14:         Adjust biases $\boldsymbol{b}_{\text{up}}, \boldsymbol{b}_{\text{down}}$ accordingly
15:     **end for**
16:     **Compute penalty:**
17:     **for** each layer $l = 1, 2, \ldots, L$ **do**
18:         $\mathcal{D}_l \leftarrow \sup_x |F_{1,n}(x) - F_{2,\bar{n}}(x)|$ with $F_{1,n}$ and $F_{2,\bar{n}}$ being the ECDF of singular values of $\boldsymbol{W}_l^{\text{compressed}}$ and $\boldsymbol{W}_l$, respectively.        ▷ Kolmogorov-Smirnov distance
19:         $G_l \leftarrow \sum_{k=0}^{\infty} \gamma^k \mathcal{D}_{l+k+1}$               ▷ Discounted future penalty
20:     **end for**
21:     **Optimize policy learner:**
22:     Compute gradient using REINFORCE:

$$\nabla \mathbb{E}[G_l] \propto \nabla \log P(S_l | \boldsymbol{W}_l) \cdot G_l$$

23:     Update $\boldsymbol{W}_{\text{inter}}, \boldsymbol{W}_{\text{proj}}$
24: **end for**

---

## C PROOFS OF THE THEORETICAL RESULTS

### C.1 PROOF OF LEMMA 3.1

*Proof.* Suppose an LLM has $d_{\text{hidden}}$ as the hidden size and $d_{\text{intermediate}}$ as the intermediate FFN size. Suppose the decoder vocabulary is $V$. Therefore, the total number of parameters introduced in the decoder input embedding and output heads is $|V| \cdot d_{\text{hidden}}$. Each self-attention query, value and key projection matrices has $d_{\text{hidden}} \cdot d_{\text{hidden}}$ parameters and the attention output projection has $d_{\text{hidden}} \cdot d_{\text{hidden}}$ parameters. Each FFN layer has total $2d_{\text{hidden}} \cdot d_{\text{intermediate}}$ parameters (here we assume that the LLM has only two FFNs in the MLP layer, however a similar formulation is applicable for models like LLaMA-2 that uses three FFNs). Therefore, the total number of parameters in the uncompressed model with $L$ layers is

$$P_{\text{uncompressed}} = 2|V|d_{\text{hidden}} + (4d_{\text{hidden}}^2 + 2d_{\text{hidden}}d_{\text{intermediate}})L.$$

For the compressed model with compression ratio $r$, the total number of parameters without the parameter learning component is

$$P_{\text{compressed}} = 2|V|d_{\text{hidden}} + (4pd_{\text{hidden}}^2 + 2pd_{\text{hidden}}d_{\text{intermediate}})L.$$

Where $p = 1 - r$. Here, we assume that compression will take place only on one dimension of $\boldsymbol{Q}, \boldsymbol{K}, \boldsymbol{V}, \boldsymbol{O}, \text{FFN}_1$ and $\text{FFN}_2$. The parameter learning component has $pd_{\text{hidden}}^2$ parameters for each attention module and $pd_{\text{hidden}}d_{\text{intermediate}}$ parameters for each FFN layer. Therefore, the total number of parameters in the compressed model is

$$P_{\text{compressed}} = 2|V|d_{\text{hidden}} + (5pd_{\text{hidden}}^2 + 3pd_{\text{hidden}}d_{\text{intermediate}})L.$$

Therefore, to ensure $P_{\text{compressed}} < P_{\text{uncompressed}}$ we must ensure $5pd_{\text{hidden}}^2 + 3pd_{\text{hidden}}d_{\text{intermediate}} < 4d_{\text{hidden}}^2 + 2d_{\text{hidden}}d_{\text{intermediate}}$, *i.e.,* $p < \frac{4d_{\text{hidden}}^2 + 2d_{\text{hidden}}d_{\text{intermediate}}}{5d_{\text{hidden}}^2 + 3d_{\text{hidden}}d_{\text{intermediate}}} = \frac{4d_{\text{hidden}} + 2d_{\text{intermediate}}}{5d_{\text{hidden}} + 3d_{\text{intermediate}}}$. Therefore, $r = 1 - p > 1 - \frac{4d_{\text{hidden}} + 2d_{\text{intermediate}}}{5d_{\text{hidden}} + 3d_{\text{intermediate}}} = \frac{d_{\text{hidden}} + d_{\text{intermediate}}}{5d_{\text{hidden}} + 3d_{\text{intermediate}}}$. □

### C.2 PROOF OF COROLLARY 3.3

*Proof.* Note that $\boldsymbol{W}\boldsymbol{W}^T \in \mathbb{R}^{n \times n}$ and $(\boldsymbol{W}')(\boldsymbol{W}')^T \in \mathbb{R}^{m \times m}$ are both symmetric matrices. It is easy to see that $(\boldsymbol{W}')(\boldsymbol{W}')^T$ is obtained from $\boldsymbol{W}\boldsymbol{W}^T$ by removing rows and columns of $\boldsymbol{W}\boldsymbol{W}^T$, corresponding to the row indices which have been sliced off; we'll make this formal below.

Let $\boldsymbol{A} = \boldsymbol{W}\boldsymbol{W}^T$ and $\boldsymbol{A}' = (\boldsymbol{W}')(\boldsymbol{W}')^T$. Let $S$ be the set of row indices not sliced (we assume that the set $S$ is arranged in sorted order). In particular, we have that $|S| = m$. Now, for $i \in S$, let $p(i)$ be the index of $i$ in $S$. In particular, as $i$ iterates through elements in $S$ in order, $p(i)$ iterates through the integers in the set $[m]$.

Next, consider the matrix $\boldsymbol{B} \in \mathbb{R}^{n \times m}$ given by

$$B_{ij} = \begin{cases} 1 & , \quad \text{if } i \in S \text{ and } j = p(i) \\ 0 & , \quad \text{otherwise} \end{cases}$$

We claim that $\boldsymbol{B}$ is *semi-orthogonal*, i.e $\boldsymbol{B}^T\boldsymbol{B} = \boldsymbol{I}_m$, where $\boldsymbol{I}_m \in \mathbb{R}^{m \times m}$ is the identity matrix. But this is straightforward; note that the mapping $i \mapsto p(i)$ is a one-to-one mapping. In particular, this implies that the columns of $\boldsymbol{B}$ are distinct one-hot vectors, thus forming a set of orthonormal vectors, and hence $\boldsymbol{B}^T\boldsymbol{B} = \boldsymbol{I}_m$ follows easily.

Next, we claim that

$$\boldsymbol{A}' = \boldsymbol{B}^T\boldsymbol{A}\boldsymbol{B}$$

The above equality can be derived as follows:

$$
\begin{aligned}
[(B^T)AB]_{ij} &= \sum_{l=1}^{n} (B^T A)_{il} B_{lj} \\
&= \sum_{l=1}^{n} \sum_{k=1}^{n} B_{ik}^T A_{kl} B_{lj} \\
&= \sum_{l=1}^{n} \sum_{k=1}^{n} B_{ki} A_{kl} B_{lj} \\
&= \sum_{l=1}^{n} A_{p^{-1}(i)l} B_{lj} \\
&= A_{p^{-1}(i)p^{-1}(j)} \\
&= A'_{ij}
\end{aligned}
$$

So, from Theorem 3.2, we know that the range of eigenvalues of $B^T A B$ is a subset of the range of eigenvalues values of $A$. Finally, since singular values of $W'$ and $W$ are just square roots of the eigenvalues of $B^T A B$ and $A$ respectively, our claim follows. $\qquad\square$

### C.3 The reparametrization trick in Equation 4

In this section, we motivate the formulation in Equation 4 by showing that it reparametrizes the Bernoulli distribution by using a thresholding trick. So, let $\alpha \in (0,1)$, and consider the distribution Bernoulli$(\alpha)$. Consider the following experiment: we sample $\epsilon \sim U(0,1)$, and define the following random variable:

$$
\begin{aligned}
Y &= \ln\left(\frac{\epsilon}{1-\epsilon}\right) + \ln\left(\frac{\alpha}{1-\alpha}\right) \\
&= \ln\left(\frac{\epsilon\alpha}{(1-\epsilon)(1-\alpha)}\right)
\end{aligned}
$$

In particular, if $\sigma$ is the sigmoid function, then

$$
\begin{aligned}
\sigma(Y) &= \frac{1}{1 + \frac{(1-\epsilon)(1-\alpha)}{\epsilon\alpha}} \\
&= \frac{\epsilon\alpha}{\epsilon\alpha + (1-\epsilon)(1-\alpha)}
\end{aligned}
$$

Next, consider the random variable $Z$ defined as:

$$
Z := \begin{cases} 1 &, \quad \text{if } \sigma(Y) \geq 0.5 \\ 0 &, \quad \text{otherwise} \end{cases}
$$

We claim that $Z \sim$ Bernoulli$(\alpha)$. This is straightforward; note that

$$
\begin{aligned}
\mathbb{P}[Z=1] &= \mathbb{P}[\sigma(Y) \geq 0.5] \\
&= \mathbb{P}\left[\epsilon\alpha \geq \frac{(\epsilon\alpha + (1-\epsilon)(1-\alpha))}{2}\right] \\
&= \mathbb{P}\left[\epsilon\alpha \geq (1-\epsilon)(1-\alpha)\right] \\
&= \mathbb{P}\left[\epsilon\alpha \geq 1 - \alpha - \epsilon + \epsilon\alpha\right] \\
&= \mathbb{P}[\epsilon \geq 1 - \alpha] \\
&= 1 - \mathbb{P}[\epsilon < 1 - \alpha] \\
&= 1 - (1-\alpha) \\
&= \alpha
\end{aligned}
$$

completing the proof of the claim.

## D  DATASET AND TASK DESCRIPTIONS

**Zero-shot evaluation datasets.**  PIQA (Bisk et al., 2020) is a physical common-sense reasoning dataset focusing on everyday situations with a preference for atypical solutions. Each example of the dataset provides users with instructions on how to build, craft, bake or manipulate objects using everyday materials. The common-sense reasoning task is formulated as an MCQ-based question-answering task: given a question $q$ and two possible solutions $s_1$ and $s_2$, a model/human must choose the most appropriate solution, of which exactly one is correct. The WinoGrande dataset (Sakaguchi et al., 2021) is a large-scale dataset of problems from the Winograd Schema Challenge (Levesque et al., 2012) consisting of pronoun resolution problems that are designed to be trivial for humans but complex for AI systems. The HellaSwag (Zellers et al., 2019) dataset consists of the issues describing common-sense natural language inference (NLI) tasks: given a sentence, a model/human must predict the most likely follow-up. The AI2 Reasoning Challenge (Clark et al., 2018) dataset contains natural, grade-school science question-answering problems (authored for human tests) requiring powerful knowledge and reasoning. The MMLU benchmark (Hendrycks et al., 2020) contains problems covering 57 subjects across STEM, the humanities and social science. It measures the knowledge models acquired during pre-training by evaluating them in zero-shot and few-shot settings.

**Recovery fine-tuning datasets.**  The WikiText (Merity et al., 2016) dataset is a commonly used benchmark for language modelling, consisting of articles from Wikipedia which satisfy the *Good* or *Featured* article criteria specified by editors on the platform. These articles, reviewed by humans, are considered well-written, factually correct and neutral in point of view. The dataset is available in WikiText-2 and WikiText-103; our experiments use the WikiText-2 dataset. The Penn Treebank (PTB) (Marcus et al., 1993) dataset is a large annotated corpus containing over 4.5 million words of American English. The section of the corpus corresponding to articles from the Wall Street Journal is mainly known to be used to evaluate models for sequence labelling. The Alpaca (Taori et al., 2023) dataset consists of 52000 instructions and demonstrations generated by OpenAI's `text-davinci-003` model, which is widely used for instruction tuning of language models. We use only up to 8000 samples from these datasets for recovery fine-tuning.

## E  ADDITIONAL RESULTS

### E.1  DETAILED COMPARISON WITH SLICEGPT WITH AND WITHOUT RFT

Table 10 contains a comparison of the performance of the LLaMA-2-7B and Phi-2 pruned with SliceGPT using the Alpaca calibration dataset. While LLaMA-2-7B pruned with SliceGPT exhibits better average performance for all compression ratios, Phi-2 exhibits an opposite trend, wherein `PruneNet` beats SliceGPT by a consistent margin of at least 2% for all compression ratios.

We report the results with LLaMA-2-7B with RFT on Wikitext2 and Alpaca datasets in Table 11 and Table 12, respectively. With RFT on Wikitext2 dataset, `PruneNet` achieves on average > 5% accuracy than SliceGPT. However, SliceGPT outperforms `PruneNet` when fine-tuned on the Alpaca dataset, with an average margin of 3%. As SliceGPT uses the same datasets for calibration and RFT, it typically has access to more instruction-tuning datasets than `PruneNet` allowing it to do better when fine-tuned on the Alpaca dataset. However, it is worth noticing that the average standard deviation in the performance of SliceGPT after RFT is significantly high (5.5) compared to `PruneNet` (1.5). The low standard deviation highlights the robustness of `PruneNet` when fine-tuned on different datasets for recovering the information loss during compression.

Table 13 highlights the performance of the LLaMA-2-7B model at 50% compression ratio with `PruneNet` and SliceGPT. While both the methods can regain only 60% of the performance of the original uncompressed model, `PruneNet` demonstrates 2% better than the SliceGPT, showcasing its effectiveness over the baseline, even at a very high compression rate.

In Table 14, we highlight the results with `PruneNet` and SliceGPT for different compression ratios for the LLaMA-2-13B model. The performance drops drastically for larger models like LLaMA-13B at a high compression ratio. However, the results in Table 15 highlight that after recovery fine-tuning, the compressed models regain the performance quickly and can preserve up to 84% of the original uncompressed model performance, even at a high compression rate of 30%. `PruneNet`

also outperforms SliceGPT with a margin of $2\%$, showcasing a similar trend as the smaller LLMs used previously.

## E.2 Ablation Study

We perform an ablation study to understand the importance of different components of `PruneNet`.

**Importance of Learnable Policy for Model Compression.** We conduct experiments with a random selection process, where the pruned parameter indices are chosen randomly. We highlight the results with random policy in Table 16. We observe an average $2\%$ drop with a random selection method, justifying the need to learn which parameters to compress for more effective model compression.

**Importance of Stochastic Policy.** Table 17 highlights the results with LLaMA-2-7B with deterministic and stochastic (policy learned with `PruneNet` in Equation 5) policies. In the deterministic policy, we chose only the topk parameters based on the importance metric defined in Equation 3. We observe that deterministic policy often underperforms the stochastic policy with an average margin of $4\%$. The results highlight that parameter importance alone cannot determine which parameters to compress. Preserving the spectral structure between the compressed and uncompressed models is critical to ensure minimal performance drop post-compression.

**Results with Different Reward Functions.** To further understand the effectiveness of `PruneNet` under different distance measures, we evaluate the LLaMA-2-7B model compressed using `PruneNet` with non-parametric Anderson–Darling measure of agreement. Table 18 highlights the effectiveness of `PruneNet` with both Kolmogorov-Smirnov (highlighted as KS) and Anderson–Darling (highlighted as AD) distance measures. Under both reward functions, we observe a similar performance of `PruneNet` for different compression ratios with the LLaMA-2-7B model. The results further emphasize the stability of our proposed compression method under different choice metrics.

**Importance of Policy State for Model Compression.** Table 19 highlights the zero-shot performance of the LLaMA-2-7B model compressed with `PruneNet` with policy learned on different FFN matrices. In most cases, we observe marginal differences in the result ($< 1\%$) when the policy is learned with FFN2 instead of FFN1. The observations emphasize that `PruneNet` is invariant to the choice of parameter used for learning the policy.

**Results with Pruned Self-Attention Blocks.** We highlight the pruning results for LLaMA-2-7B model with compressed self-attention layers in Table 20. The performance drop with the compressed models suggests that compressing self-attention layers intrinsically is harder than compressing FFN layers. However, around $50\%$ of the performance drop can be recovered with recovery fine-tuning.

## E.3 Additional Results with OPT and LLaMA-1 Models

Table 21 highlights the zero-shot performance of all the LLMs without RFT. OPT-125M can retain nearly 96% of its original performance even at 25% compression, suggesting strong stability under compression. Larger OPT models are more sensitive to different compression ratios where the performance drops by $8\%$ at higher compression. Table 22 highlights the zero-shot performance of the LLMs after being compressed by SliceGPT. An one-sided Kolmogorov-Smirnov (KS) test suggests ($p$-value $< 0.05$) that the performance drop exhibited by `PruneNet` is significantly lower than that of SliceGPT across all the LLMs.

Table 23 reports the zero-shot performance on multitask language understanding (MMLU) tasks. The OPT family models show a steady performance across all compression ratios. At 10% compression, the average MMLU accuracy score is 23.4 with the OPT-6.7B model, and it maintains a comparable performance of 24.5 even at 30% compression, indicating good resilience. Similar performance stability can also be observed with OPT-125M and OPT-2.7B models. However, LLaMA and Phi model performances drop significantly with higher compression ratios. However, the compressed models show strong performance on selected tasks like College Computer Science, College Mathematics, Formal Logic, and Global Facts, even outperforming the uncompressed models.

| Model | Comp. Ratio | Method | PIQA | WinoGrande | HellaSwag | ARC-e | ARC-c | Avg. |
|---|---|---|---|---|---|---|---|---|
| LLaMA-2-7B | 0% | Dense | 79.11 | 69.06 | 75.99 | 74.58 | 46.25 | 69.00 |
| | 20% | SliceGPT | 76.60 | 65.51 | 65.20 | 69.99 | 41.21 | 63.68 |
| | | PruneNet | 75.53 | 65.51 | 66.43 | 63.80 | 37.29 | 61.71 |
| | 25% | SliceGPT | 74.21 | 64.01 | 60.55 | 66.88 | 38.91 | 60.91 |
| | | PruneNet | 72.09 | 62.43 | 62.33 | 60.14 | 36.18 | 58.63 |
| | 30% | SliceGPT | 72.25 | 59.93 | 55.86 | 63.93 | 37.80 | 57.93 |
| | | PruneNet | 71.11 | 61.09 | 58.30 | 53.20 | 33.53 | 55.45 |
| Phi-2 | 0% | Dense | 79.11 | 75.77 | 73.83 | 78.32 | 54.18 | 72.24 |
| | 20% | SliceGPT | 76.17 | 68.75 | 61.95 | 72.18 | 45.48 | 64.90 |
| | | PruneNet | 74.37 | 70.80 | 65.53 | 74.71 | 47.53 | 66.59 |
| | 25% | SliceGPT | 75.68 | 64.88 | 58.19 | 70.41 | 43.43 | 62.52 |
| | | PruneNet | 74.37 | 68.98 | 62.18 | 70.54 | 44.45 | 64.10 |
| | 30% | SliceGPT | 74.05 | 62.12 | 53.31 | 67.26 | 39.42 | 59.23 |
| | | PruneNet | 72.80 | 67.48 | 56.80 | 67.55 | 40.61 | 61.05 |

Table 10: LLaMA-2-7B results compressed with `PruneNet` without RFT. SliceGPT uses Alpaca dataset for calibration.

| Comp. Ratio | Method | PIQA | WinoGrande | HellaSwag | ARC-e | ARC-c | Avg. |
|---|---|---|---|---|---|---|---|
| 0% | Dense | 79.11 | 69.06 | 75.99 | 74.58 | 46.25 | 69.00 |
| 20% | SliceGPT | 69.86 | 64.72 | 61.07 | 54.25 | 36.43 | 57.27 |
| | PruneNet | 74.76 | 66.22 | 69.38 | 65.61 | 39.25 | 63.04 |
| 25% | SliceGPT | 69.26 | 64.96 | 58.65 | 52.36 | 35.75 | 56.20 |
| | PruneNet | 74.37 | 66.46 | 65.71 | 60.82 | 36.60 | 60.79 |
| 30% | SliceGPT | 67.41 | 63.22 | 55.65 | 50.76 | 34.13 | 54.23 |
| | PruneNet | 73.01 | 63.46 | 63.21 | 60.14 | 35.92 | 59.15 |

Table 11: LLaMA-2-7B results compressed with `PruneNet` and SliceGPT with recovery fine-tuning on WikiText2 dataset.

| Comp. Ratio | Method | PIQA | WinoGrande | HellaSwag | ARC-e | ARC-c | Avg. |
|---|---|---|---|---|---|---|---|
| 0% | Dense | 79.11 | 69.06 | 75.99 | 74.58 | 46.25 | 69.00 |
| 20% | SliceGPT | 76.55 | 65.59 | 68.26 | 71.84 | 45.05 | 65.46 |
| | PruneNet | 72.73 | 62.25 | 66.45 | 61.52 | 42.15 | 61.02 |
| 25% | SliceGPT | 75.79 | 63.22 | 65.12 | 68.22 | 42.83 | 63.04 |
| | PruneNet | 75.79 | 62.35 | 65.48 | 60.94 | 39.16 | 60.74 |
| 30% | SliceGPT | 74.59 | 61.64 | 63.06 | 66.54 | 40.87 | 61.34 |
| | PruneNet | 72.14 | 62.75 | 62.38 | 55.43 | 37.03 | 57.95 |

Table 12: LLaMA-2-7B results compressed with `PruneNet` and SliceGPT with recovery fine-tuning on Alpaca dataset.

| Method | PIQA | WinoGrande | HellaSwag | ARC-e | ARC-c | Avg. |
|---|---|---|---|---|---|---|
| Dense | 79.11 | 69.06 | 75.99 | 74.58 | 46.25 | 69.00 |
| SliceGPT | 53.97 | 53.04 | 32.65 | 34.76 | 23.72 | 39.63 |
| PruneNet | 59.68 | 52.09 | 35.21 | 34.89 | 25.43 | 41.46 |

Table 13: LLaMA-2-7B results compressed with `PruneNet` without RFT at 50% compression ratio.

| Comp. Ratio | Method | PIQA | WinoGrande | HellaSwag | ARC-e | ARC-c | Avg. |
|---|---|---|---|---|---|---|---|
| 0% | Dense | 80.47 | 72.22 | 79.39 | 77.48 | 49.23 | 71.76 |
| 20% | SliceGPT | 71.87 | 69.38 | 63.04 | 69.87 | 43.09 | 63.45 |
| 20% | PruneNet | 77.15 | 66.38 | 72.90 | 70.50 | 41.81 | 65.75 |
| 25% | SliceGPT | 68.55 | 67.48 | 58.1 | 62.5 | 37.88 | 58.90 |
| 25% | PruneNet | 70.89 | 62.43 | 58.67 | 58.63 | 34.04 | 56.93 |
| 30% | SliceGPT | 66.1 | 65.11 | 52.69 | 56.82 | 35.07 | 55.16 |
| 30% | PruneNet | 61.92 | 56.99 | 35.65 | 46.34 | 28.33 | 45.87 |

Table 14: LLaMA-2-13B results compressed with `PruneNet` and SliceGPT without RFT.

| Comp. Ratio | Method | PIQA | WinoGrande | HellaSwag | ARC-e | ARC-c | Avg. |
|---|---|---|---|---|---|---|---|
| 0% | Dense | 80.47 | 72.22 | 79.39 | 77.48 | 49.23 | 71.76 |
| 20% | SliceGPT | 74.10 | 68.51 | 66.94 | 70.54 | 43.77 | 64.77 |
| 20% | PruneNet | 76.22 | 68.43 | 70.72 | 66.88 | 42.83 | 65.02 |
| 25% | SliceGPT | 71.27 | 68.98 | 64.12 | 63.76 | 40.87 | 61.8 |
| 25% | PruneNet | 76.93 | 64.80 | 70.44 | 66.96 | 40.36 | 63.90 |
| 30% | SliceGPT | 69.64 | 66.85 | 59.93 | 59.55 | 38.65 | 58.92 |
| 30% | PruneNet | 73.45 | 65.59 | 64.5 | 60.73 | 38.57 | 60.57 |

Table 15: LLaMA-2-13B results compressed with `PruneNet` and SliceGPT with RFT on Wiki-Text2 dataset.

| Comp. Ratio | Selection | PIQA | WinoGrande | HellaSwag | ARC-e | ARC-c | Average |
|---|---|---|---|---|---|---|---|
| 20% | Policy-based | 75.3 | 65.5 | 66.43 | 63.8 | 37.29 | 61.66 |
| | Random | 72.36 | 63.14 | 61.18 | 60.31 | 36.52 | 58.70 |
| 25% | Policy-based | 72.09 | 62.43 | 62.33 | 60.14 | 36.18 | 58.63 |
| | Random | 70.13 | 60.38 | 58.38 | 56.27 | 35.67 | 56.20 |
| 30% | Policy-based | 71.13 | 61.09 | 58.30 | 53.20 | 33.53 | 55.45 |
| | Random | 73.13 | 59.35 | 55.15 | 49.83 | 31.66 | 53.90 |

Table 16: Effect of learnable policy for compressing an LLaMA-2-7B model.

| Comp. Ratio | Policy | PIQA | WinoGrande | HellaSwag | ARC-e | ARC-c | Average |
|---|---|---|---|---|---|---|---|
| 20% | Stochastic | 75.3 | 65.5 | 66.43 | 63.8 | 37.29 | 61.66 |
| | Deterministic | 72.91 | 61.64 | 61.05 | 56.69 | 36.52 | 57.76 |
| 25% | Stochastic | 72.09 | 62.43 | 62.33 | 60.14 | 36.18 | 58.63 |
| | Deterministic | 69.97 | 59.27 | 59.39 | 56.69 | 33.53 | 55.77 |
| 30% | Stochastic | 71.13 | 61.09 | 58.30 | 53.20 | 33.53 | 55.45 |
| | Deterministic | 69.64 | 58.25 | 54.45 | 54.97 | 31.23 | 53.71 |

Table 17: Effect of stochastic policy on compressed LLaMA-2-7B model.

| Comp. Ratio | Reward Function | PIQA | WinoGrande | HellaSwag | ARC-e | ARC-c | Average |
|---|---|---|---|---|---|---|---|
| 20% | KS | 75.3 | 65.5 | 66.43 | 63.8 | 37.29 | 61.66 |
| | AD | 73.01 | 63.3 | 65.7 | 60.4 | 37.46 | 59.97 |
| 25% | KS | 72.09 | 62.43 | 62.33 | 60.14 | 36.18 | 58.63 |
| | AD | 73.88 | 61.17 | 63.98 | 61.62 | 35.84 | 59.30 |
| 30% | KS | 71.13 | 61.09 | 58.30 | 53.20 | 33.53 | 55.45 |
| | AD | 72.13 | 61.88 | 60.18 | 58.00 | 33.62 | 57.16 |

Table 18: Zero-shot performance of LLaMA-2-7B compressed with `PruneNet` with different reward functions.

| Comp. Ratio | Layer | PIQA | WinoGrande | HellaSwag | ARC-e | ARC-c | Avg. |
|---|---|---|---|---|---|---|---|
| 20% | FFN1 | 75.30 | 65.50 | 66.43 | 63.80 | 37.29 | 61.66 |
| | FFN2 | 74.81 | 66.93 | 67.38 | 61.24 | 36.86 | 61.44 |
| 25% | FFN1 | 72.09 | 62.43 | 62.33 | 60.14 | 36.18 | 58.63 |
| | FFN2 | 70.13 | 57.30 | 59.98 | 55.51 | 34.22 | 55.43 |
| 30% | FFN1 | 71.11 | 61.09 | 58.30 | 53.20 | 33.53 | 55.45 |
| | FFN2 | 72.20 | 61.56 | 60.01 | 54.12 | 33.70 | 56.32 |

Table 19: A comparison of using FFN1 vs FFN2 modules for learning policy with `PruneNet`.

| Comp. Ratio | RFT | PIQA | WinoGrande | HellaSwag | ARC-e | ARC-c | Avg. |
|---|---|---|---|---|---|---|---|
| 20% | None | 50.11 | 52.49 | 26.24 | 27.31 | 28.75 | 36.98 |
| 20% | WikiText2 | 71.98 | 56.83 | 53.04 | 55.13 | 33.45 | 54.09 |
| 25% | None | 50.11 | 50.28 | 26.25 | 25.99 | 28.16 | 36.16 |
| 25% | WikiText2 | 69.42 | 54.93 | 40.24 | 49.87 | 32.68 | 49.43 |
| 30% | None | 50.59 | 49.57 | 26.58 | 26.56 | 26.54 | 35.97 |
| 30% | WikiText2 | 62.46 | 52.01 | 47.53 | 38.55 | 28.75 | 45.86 |

Table 20: Results with LLaMA-2-7B model with pruned self-attention modules.

| Model | Comp. Ratio | PIQA | WinoGrande | HellaSwag | ARC-e | ARC-c | Average |
|---|---|---|---|---|---|---|---|
| OPT-125M | 0% | 61.97 | 50.28 | 31.35 | 39.90 | 22.78 | 41.26 |
| | 20% | 58.49 | 52.49 | 30.43 | 37.46 | 21.50 | 40.07 |
| | 25% | 59.41 | 51.78 | 30.19 | 35.31 | 21.93 | 39.72 |
| | 30% | 58.75 | 45.88 | 29.03 | 34.15 | 20.79 | 37.72 |
| OPT-2.7B | 0% | 74.81 | 61.09 | 60.63 | 54.25 | 31.31 | 56.42 |
| | 20% | 70.02 | 58.33 | 50.52 | 46.30 | 27.73 | 50.58 |
| | 25% | 68.88 | 57.46 | 47.91 | 45.83 | 27.82 | 49.58 |
| | 30% | 66.21 | 56.99 | 46.61 | 43.56 | 27.05 | 48.08 |
| OPT-6.7B | 0% | 76.55 | 65.35 | 67.22 | 60.06 | 34.81 | 60.80 |
| | 20% | 72.80 | 61.64 | 59.08 | 49.49 | 31.06 | 54.81 |
| | 25% | 72.69 | 60.38 | 57.01 | 48.61 | 30.38 | 53.81 |
| | 30% | 70.84 | 58.25 | 55.50 | 46.93 | 29.95 | 52.29 |
| LLaMA-2-7B | 0% | 79.11 | 69.30 | 76.00 | 74.62 | 46.25 | 69.06 |
| | 20% | 75.30 | 65.51 | 66.43 | 63.80 | 37.29 | 61.67 |
| | 25% | 72.09 | 62.43 | 62.33 | 60.14 | 36.18 | 58.63 |
| | 30% | 71.11 | 61.09 | 58.30 | 53.20 | 33.53 | 55.45 |
| Phi-2 | 0% | 79.16 | 76.01 | 73.84 | 78.24 | 54.01 | 72.25 |
| | 20% | 74.37 | 70.80 | 65.53 | 74.71 | 47.53 | 66.59 |
| | 25% | 74.37 | 68.98 | 62.18 | 70.54 | 44.45 | 64.10 |
| | 30% | 72.80 | 67.48 | 56.80 | 67.55 | 40.61 | 61.05 |
| LLaMA-1-7B | 0% | 78.24 | 67.41 | 65.78 | 67.38 | 38.14 | 63.39 |
| | 20% | 75.51 | 62.12 | 65.40 | 64.65 | 36.52 | 60.82 |
| | 25% | 72.18 | 60.40 | 59.55 | 61.93 | 32.37 | 57.29 |
| | 30% | 69.91 | 59.27 | 57.19 | 52.23 | 30.80 | 53.88 |

Table 21: Zero-shot performance of LLMs compressed with `PruneNet` without recovery fine-tuning.

| Model | Comp. Ratio | PIQA | WinoGrande | HellaSwag | ARC-e | ARC-c | Average |
|---|---|---|---|---|---|---|---|
| OPT-125M | 20% | 55.19 | 46.57 | 27.23 | 36.86 | 21.24 | 37.42 |
| | 25% | 53.35 | 44.10 | 25.72 | 35.07 | 20.99 | 35.85 |
| | 30% | 52.79 | 43.57 | 24.91 | 35.13 | 19.35 | 35.15 |
| OPT-2.7B | 20% | 68.23 | 57.93 | 51.38 | 51.81 | 28.5 | 51.57 |
| | 25% | 65.29 | 57.22 | 47.85 | 49.79 | 27.99 | 49.63 |
| | 30% | 62.35 | 57.22 | 44.18 | 46.72 | 27.05 | 47.50 |
| OPT-6.7B | 20% | 72.74 | 61.09 | 61.04 | 55.89 | 30.8 | 56.31 |
| | 25% | 70.35 | 60.62 | 58.15 | 52.78 | 29.52 | 54.28 |
| | 30% | 68.61 | 60.69 | 54.56 | 52.15 | 29.01 | 53.00 |
| LLAMA-2-7B | 20% | 69.42 | 65.11 | 59.04 | 59.76 | 37.54 | 58.18 |
| | 25% | 66.87 | 63.38 | 54.16 | 58.46 | 34.56 | 55.48 |
| | 30% | 63.55 | 61.33 | 49.62 | 51.77 | 31.23 | 51.50 |
| Phi-2 | 20% | 71.87 | 67.80 | 57.76 | 58.00 | 35.32 | 58.15 |
| | 25% | 69.21 | 65.35 | 52.40 | 53.70 | 31.66 | 54.46 |
| | 30% | 65.94 | 63.14 | 47.56 | 53.03 | 30.29 | 51.99 |
| LLAMA-1-7B | 20% | 72.44 | 62.45 | 61.30 | 63.59 | 35.09 | 58.97 |
| | 25% | 70.19 | 61.60 | 58.58 | 60.08 | 33.56 | 56.80 |
| | 30% | 68.57 | 60.55 | 56.26 | 49.14 | 31.95 | 53.33 |

Table 22: Zero-shot performance of LLMs compressed with SliceGPT without recovery fine-tuning.

| Model | Comp. Ratio | Abs. Alg. | Bus. Eth. | Clg. CS. | Clg. Math | Con. Phy. | Frm. Log. | Glb. Fct. | ML. | Misc. | Phil. | Avg. |
|---|---|---|---|---|---|---|---|---|---|---|---|---|
| OPT-125M | 0% | 21 | 31 | 24 | 20 | 27 | 29 | 18 | 29 | 23 | 19 | 24 |
| | 20% | 21 | 30 | 26 | 21 | 28 | 29 | 18 | 33 | 23 | 19 | 25 |
| | 25% | 22 | 30 | 25 | 21 | 26 | 29 | 18 | 31 | 23 | 18 | 24 |
| | 30% | 20 | 28 | 25 | 21 | 25 | 27 | 17 | 31 | 21 | 17 | 23 |
| OPT-2.7B | 0% | 26 | 26 | 34 | 28 | 23 | 21 | 29 | 25 | 25 | 31 | 27 |
| | 20% | 21 | 23 | 28 | 23 | 24 | 29 | 25 | 30 | 24 | 20 | 25 |
| | 25% | 22 | 28 | 27 | 22 | 27 | 29 | 17 | 32 | 24 | 19 | 25 |
| | 30% | 22 | 30 | 26 | 21 | 26 | 29 | 18 | 31 | 24 | 19 | 25 |
| OPT-6.7B | 0% | 24 | 24 | 33 | 29 | 22 | 20 | 27 | 27 | 26 | 20 | 25 |
| | 20% | 22 | 30 | 26 | 21 | 27 | 28 | 18 | 31 | 23 | 19 | 24 |
| | 25% | 22 | 30 | 27 | 21 | 26 | 28 | 18 | 34 | 23 | 19 | 25 |
| | 30% | 22 | 30 | 25 | 21 | 26 | 27 | 18 | 33 | 24 | 19 | 24 |
| LLaMA-2 | 0% | 24 | 46 | 37 | 31 | 39 | 30 | 25 | 40 | 55 | 51 | 38 |
| | 20% | 29 | 36 | 33 | 32 | 31 | 21 | 36 | 32 | 33 | 33 | 32 |
| | 25% | 28 | 30 | 26 | 29 | 26 | 29 | 22 | 31 | 27 | 30 | 28 |
| | 30% | 25 | 34 | 37 | 31 | 27 | 36 | 24 | 30 | 26 | 24 | 29 |
| Phi-2 | 0% | 30 | 58 | 45 | 44 | 47 | 33 | 37 | 49 | 69 | 57 | 47 |
| | 20% | 27 | 56 | 31 | 34 | 38 | 27 | 30 | 30 | 58 | 48 | 38 |
| | 25% | 20 | 48 | 29 | 33 | 42 | 28 | 33 | 37 | 52 | 50 | 37 |
| | 30% | 19 | 35 | 42 | 25 | 25 | 25 | 31 | 25 | 44 | 32 | 30 |
| LLaMA-1 | 0% | 22 | 35 | 38 | 35 | 30 | 32 | 29 | 23 | 35 | 36 | 32 |
| | 20% | 21 | 30 | 27 | 22 | 26 | 27 | 20 | 31 | 26 | 20 | 25 |
| | 25% | 20 | 27 | 30 | 26 | 25 | 28 | 19 | 21 | 28 | 29 | 25 |
| | 30% | 23 | 20 | 37 | 34 | 20 | 35 | 15 | 16 | 25 | 24 | 25 |

Table 23: Zero-shot performance of compressed LLMs on MMLU (Massive Multitask Language Understanding) tasks without RFT. Task descriptions: **Abs. Alg:** Abstract Algebra, **Bus. Eth. :** Business Ethics, **Clg. CS.:** College Computer Science, **Clg. Math:** College Mathematics, **Con. Phy.:** Conceptual Physics, **Frm. Log.:** Formal Logic, **Glb. Fct.:** Global Facts, **ML.:** Machine Learning, **Misc.:** Miscellaneous, **Phil.:** Philosophy.

