# OpenReview forum: "You Only Prune Once: Designing Calibration-Free Model Compression With Policy Learning"
_ICLR.cc/2025/Conference — ICLR 2025 Poster_

### Official Review · Reviewer_onBH · 2024-10-29

**Soundness:** 4
**Presentation:** 4
**Contribution:** 3
**Rating:** 6
**Confidence:** 4

**Summary:**

This paper describes a technique -- PruneNet that avoids the usage of calibration data for structured pruning. They instead reformulate it as a policy learning process via (prun)ing-object(e)d (net)work. They does this by first making a corollary that the range of eigenvalues of the compressed model weights is a subset of the original model weights (corollary 3.3). Following that, the key idea is to identify (using an MLP/policy learner) and remove rows/columns in such a way that the distribution (and distance using Kolmogorov Smirnoff (KS) Distance) of the eigenvalues for original and compressed weights remain the same.

The method seems to achieve robust results across various models/datasets and compression ratios without a calibration dataset.

**Strengths:**

1. The removal of dependence on calibration dataset while pruning. As noted by authors in L115, calibration dataset has an impact on performance of models, so it has important effects for applications.
2. The intrinsic model compression is an interesting observation. This helps practitioners understand the dependency of compression methods (whether method X is adding parameters or not) with respect to the compression ratios
3. The reusability of policy learner i.e train a learner (MLP) for one compression ratio and use it for another ratio as the authors have noted down is a huge strength (L412)
4. Very well written with clear presentation.

I believe the paper would be a great addition with some clarifications needed in design space and experiments.

**Weaknesses:**

I don't have a list of "weakness" i.e substantial missing components. Please refer the questions section for potential ideas and clarifications needed.

**Questions:**

***Model design***

1. The sampling is done on FFN1 and complementary columns are chosen in FFN2 to match the dimensions. This makes sense logically/intuitively but did the authors experiment with sampling FFN2 separately and/or any other design choices. Does the indices match the intuitive selection. Or does the authors have an explanation on why this choice is better compared to other intuitive selections?
2. Did the authors perform experiments on usage of other penalty functions (KS)? That can be a good ablation study to have.
3. L292: The assumption of discounted penalty works if the LLM has higher singular values at later layers. Has there been any (previous) analysis on existing LLMs whether that’s the scenario? A reference would be great if it exists. If not, having plots similar to Figure 1 for other LLMs at appendix would be great.

***Experiments***

4. L361: Are the results for the one-sided test and additional details available? A reference would be great.
5. L375 (RFT): Can the authors add more insights on why RFT on model X might reduce the performance?
6. L375 (RFT): Can the RFT be done on SliceGPT as well (i.e Table 3 looking similar to Table 2). That will help to understand whether RFT helps SliceGPT more (>1.5%) and PruneNet less (marginal 1.5%) or less for both methods (in either case, it will be a useful finding to understand the effects of RFT)

***Suggestion for additional experiment on design choice***

1. Can this policy be extended to attention matrices as well? From my understanding, the key idea seems to be applicable on any matrix, so this experiment will be super helpful to understand the design space more clearly? There has been some studies [1] done to understand the effects of compression on different modules in a transformer, so the authors might take some inspiration to understand the effects on compressing on different modules.
2. Can the authors perform an experiment with >40% compression ratio? The performance drops most likely with higher compression ratios [1, 2, 3] but it would be great to know how much the PruneNet can achieve for reasonable compression.

***Possible relevant citation***
1. The Cost of Compression: Investigating the Impact of Compression on Parametric Knowledge in Language Models - https://arxiv.org/abs/2312.00960
2. Are sixteen heads really better than one? - https://arxiv.org/abs/1905.10650
3. Compressing bert: Studying the effects of weight pruning on transfer learning - https://arxiv.org/abs/2002.08307

***Format***

1. Line 37 - Can be more descriptive about the model sizes rather than a simple statement (eg: Llama 2 can be of different sizes)
2. Can the authors give clear explanations on different between Effective Sparsity and Sparsity (maybe in Appendix if possible)

---

> ### Author Response · Authors · 2024-11-19
> **Response to Reviewer onBH Comments - Part 1**
>
> We thank the reviewer for the valuable feedback. We address the concerns raised. All the changes in the main manuscript are highlighted with blue color.
>
> **A comparison of pruning on FFN1 vs FFN2 layers.** Table 1 (Table 19 of the updated paper) highlights the zero-shot performance of the LLaMA-2-7B model compressed with PruneNet with policy learned on different FFN matrices. In most cases, we observe marginal differences in the result ($< 1$%) when the policy is learned with FFN2 instead of FFN1. The observations emphasize that PruneNet is invariant to the choice of parameter used for learning the policy.
>
> ### Table 1. A comparison of FFN1 vs FFN2 layers within the PruneNet framework for Llama-2-7B.
>
> | Compression Ratio |   Layer    |  PIQA   | WinoGrande | HellaSwag |  ARC-e  |  ARC-c  |   Avg.   |
> |       -----       | ---------- | ------- |  -------   |  -------  | ------- | ------- | -------- |
> |        20%        |    FFN1    |  75.30  |   65.50    |   66.43   |  63.80  |  37.29  |  61.66   |
> |        20%        |    FFN2    |  74.81  |   66.93    |   67.38   |  61.24  |  36.86  |  61.44   |
> |        25%        |    FFN1    |  72.09  |   62.43    |   62.33   |  60.14  |  36.18  |  58.63   |
> |        25%        |    FFN2    |  70.13  |   57.30    |   59.98   |  55.51  |  34.22  |  55.43   |
> |        30%        |    FFN1    |  71.11  |   61.09    |   58.30   |  53.20  |  33.53  |  55.45   |
> |        30%        |    FFN2    |  72.20  |   61.56    |   60.01   |  54.12  |  33.70  |  56.32   |
>
> **Usage of reward functions other than the KS distance.** It is important to notice that compression reduces the cardinality of the spectrum. Therefore, the traditional pointwise distance measures (like Euclidean/Cosine/Frobenius) are not applicable for comparing the distance between the spectrum structure of uncompressed and compressed models. Therefore, we resort to probabilistic distance measures that can capture the shift in distribution structures of the compressed model.
> To further understand the effectiveness of PruneNet under different distance measures, we evaluate the LLaMA-2-7B model compressed using PruneNet with non-parametric Anderson–Darling measure of agreement. Table 2 (Table 18 of the updated paper) highlights the effectiveness of PruneNet with both Kolmogorov-Smirnov (highlighted as KS) and  Anderson–Darling (highlighted as AD) distance measures. Under both reward functions, we observe a similar performance of PruneNet for different compression ratios with the LLaMA-2-7B model. The results further emphasize the stability of our proposed compression method under different choice metrics.
>
> ### Table 2. Zero-shot performance of Llama-2-7B compressed with PruneNet with different reward functions.
>
> | Compression Ratio | Reward Function |  PIQA   | WinoGrande | HellaSwag |  ARC-e  |  ARC-c  |   Avg.   |
> |       -----       |   ----------    | ------- |  -------   |  -------  | ------- | ------- | -------- |
> |        20%        |       KS        |  75.30  |   65.50    |   66.43   |  63.80  |  37.20  |  61.66   |
> |        20%        |       AD        |  73.01  |   63.30    |   65.70   |  60.40  |  37.46  |  59.97   |
> |        25%        |       KS        |  72.09  |   62.43    |   62.33   |  60.14  |  36.18  |  58.63   |
> |        25%        |       AD        |  73.88  |   61.17    |   63.98   |  61.62  |  35.84  |  59.30   |
> |        30%        |       KS        |  71.13  |   61.09    |   58.30   |  53.20  |  33.53  |  55.45   |
> |        30%        |       AD        |  72.13  |   61.88    |   60.18   |  58.00  |  33.62  |  57.16   |
>
> **Motivation behind the usage of discounted rewards and singular values of deeper layers.** Weight matrices of LLMs have been known to exhibit a variation in the distribution of singular values across layers [1]. To further verify our observations from Figure 1 of the main text, we conduct a similar study on the FFN1 matrices of the OPT and Phi-2 models. We plot the cumulative distribution of the singular values of these matrices and observe consistent behaviour across all three model architectures. Figures 5a and 5b in the updated paper highlight the spectrum of the FFN1 module at different layers for Phi-2 and OPT-2.7B models, respectively. This further strengthens our assumption of using discounted rewards in our policy learning approach.

---

> ### Author Response · Authors · 2024-11-19
> **Response to Reviewer onBH Comments - Part 2**
>
> **Detailed results of the one-sided KS test.** For LLaMA-2-7B and Phi-2 models, we calculate the recovered performance ($\frac{\text{Compressed model performance}}{\text{Dense model performance}}$) for different compression ratios with PruneNet and SliceGPT. We conducted a one-sided KS test with the null hypothesis as SliceGPT recovered performance higher than PruneNet recovered performance for different compression ratios. For a compression ratio of $20$%, we obtained the test statistics value of 0.5 and $p$-value of $0.043$, suggesting that we reject the null hypothesis. Similarly, for compression ratio $25$% and $30$% we obtain statistics $0.5$ ($p$-value $0.043$) and $0.6$ ($p$-value $0.026$), respectively. With these statistical results, we conclude that the performance recovered with PruneNet is higher than SliceGPT with statistical significance.
>
> **Why does RFT *reduce* performance of models like Phi2?** Although interpreting the performance of a model compression method requires much detailed analysis, which is much out of scope for our current study, we can make certain educated guesses to understand the behaviours of different LLMs under compression. One possible reason behind the inferior performance of Phi-2 under RFT could be attributed to its pre-training objective. Phi-2 is pre-trained on a vast amount of synthetic texts [2], which allows it to perform reasonably well on common sense and logical reasoning tasks (the tasks used in our work) even without fine-tuning. Therefore, fine-tuning on additional datasets like WikiText2 or Alpaca could hurt the model's performance. Table 2 of the paper shows that the post-compression Phi-2 model demonstrates higher performance recovery than any other LLM, indicating its robustness on complex reasoning tasks, even after compression. Unlike SliceGPT, PruneNet preserves the original knowledge base of LLMs by preserving spectral structures. Therefore, additional fine-tuning can distort the reasoning abilities of the compressed LLM, adversely affecting its zero-shot performance.
>
> **A comparison of PruneNet with SliceGPT with RFT.** We report the results with LLaMA-2-7B with RFT on Wikitext2 and Alpaca datasets in Table 3 (Table 11 of the updated paper) and Table 4 (Table 12 of the updated paper), respectively. With RFT on the Wikitext2 dataset, PruneNet achieves, on average, $>5$% accuracy than SliceGPT. However, SliceGPT outperforms PruneNet when fine-tuned on the Alpaca dataset, with an average margin of $3$%. As SliceGPT uses the same datasets for calibration and RFT, it typically has access to more instruction-tuning datasets than PruneNet, allowing it to do better when fine-tuned on the Alpaca dataset. However, it is worth noticing that the average standard deviation in the performance of SliceGPT after RFT is significantly high ($5.5$) compared to PruneNet ($1.5$). The low standard deviation highlights the robustness of PruneNet when fine-tuned on different datasets to recover information loss during compression.
>
> ### Table 3. Results with RFT on WikiText2 on Llama-2-7B
>
> | Compression Ratio |   Method   |  PIQA   | WinoGrande | HellaSwag |  ARC-e  |  ARC-c  |   Avg.   |
> |       -----       | ---------- | ------- |  -------   |  -------  | ------- | ------- | -------- |
> |        0%         |   Dense    |  79.11  |   69.06    |   75.99   |  74.58  |  46.25  |  69.00   |
> |        20%        |  SliceGPT  |  69.86  |   64.72    |   61.07   |  54.25  |  36.73  |  57.27   |
> |        20%        |  PruneNet  |  74.76  |   66.22    |   69.38   |  65.61  |  39.25  |  63.04   |
> |        25%        |  SliceGPT  |  69.26  |   64.96    |   58.65   |  52.36  |  35.75  |  56.20   |
> |        25%        |  PruneNet  |  74.37  |   66.46    |   65.71   |  60.82  |  36.60  |  60.79   |
> |        30%        |  SliceGPT  |  67.41  |   63.22    |   55.65   |  50.76  |  34.13  |  54.23   |
> |        30%        |  PruneNet  |  73.01  |   63.46    |   63.21   |  60.14  |  35.92  |  59.15   |
>
> ### Table 4. Results with RFT on Alpaca on Llama-2-7B
>
> | Compression Ratio |   Method   |  PIQA   | WinoGrande | HellaSwag |  ARC-e  |  ARC-c  |   Avg.   |
> |       -----       | ---------- | ------- |  -------   |  -------  | ------- | ------- | -------- |
> |        0%         |   Dense    |  79.11  |   69.06    |   75.99   |  74.58  |  46.25  |  69.00   |
> |        20%        |  SliceGPT  |  76.55  |   65.59    |   68.26   |  71.84  |  45.05  |  65.46   |
> |        20%        |  PruneNet  |  72.73  |   62.25    |   66.45   |  61.52  |  42.15  |  61.02   |
> |        25%        |  SliceGPT  |  75.79  |   63.22    |   65.12   |  68.22  |  42.83  |  63.04   |
> |        25%        |  PruneNet  |  75.79  |   62.35    |   65.48  |  60.94  |  39.16  |  60.74   |
> |        30%        |  SliceGPT  |  74.59  |   61.64    |   63.06   |  66.54  |  40.87  |  61.34   |
> |        30%        |  PruneNet  |  72.14  |   62.75    |   62.38   |  55.43  |  37.03  |  57.95   |

---

> ### Author Response · Authors · 2024-11-19
> **Response to Reviewer onBH Comments - Part 3**
>
> **An implementation of policy-based pruning for attention layers.** We carry out a similar pruning strategy on attention layers, wherein we learn importance scores of the rows for the *output matrix* of the attention layer, and correspondingly slice off the key, query and value matrices to keep the output dimension consistent. We highlight the pruning results for the LLaMA-2-7B model with compressed self-attention layers in Table 5 (Table 20 of the updated paper). The performance drop with the compressed models suggests that compressing self-attention layers intrinsically is harder than compressing FFN layers. However, around $50$% of the performance drop can be recovered with recovery fine-tuning.
>
> ### Table 5. Results with Llama-2-7B with pruned self-attention modules.
>
> | Compression Ratio |    RFT     |  PIQA   | WinoGrande | HellaSwag |  ARC-e  |  ARC-c  |   Avg.   |
> |       -----       | ---------- | ------- |  -------   |  -------  | ------- | ------- | -------- |
> |        20%        |    None    |  50.11  |   52.49    |   26.24   |  27.31  |  28.75  |  36.98   |
> |        20%        | WikiText2  |  71.98  |   56.83    |   53.04   |  55.13  |  33.45  |  54.09   |
> |        25%        |    None    |  50.11  |   50.28    |   26.25   |  25.99  |  28.16  |  36.16   |
> |        25%        | WikiText2  |  69.42  |   54.93    |   40.24   |  49.87  |  32.68  |  49.43   |
> |        30%        |    None    |  50.59  |   49.57    |   26.58   |  26.56  |  26.54  |  35.97   |
> |        30%        | WikiText2  |  62.46  |   52.01    |   47.53   |  38.55  |  28.75  |  45.86   |
>
> **Performance drops for PruneNet with high compression ratios ($>40$%).** Table 6 (Table 13 of the updated paper) highlights the performance of the LLaMA-2-7B model at $50$% compression ratio with PruneNet and SliceGPT. While both methods can regain only $60$% of the performance of the original uncompressed model, PruneNet demonstrates $2$% better than the SliceGPT, showcasing its effectiveness over the baseline, even at a very high compression rate.
>
> ### Table 6. Results with 50% compression ratio on Llama-2-7B
>
> |  Method  |  PIQA   | WinoGrande | HellaSwag |  ARC-e  |  ARC-c  |   Avg.   |
> |  -----   | ------- |  -------   |  -------  | ------- | ------- | -------- |
> |  Dense   |  79.11  |   69.06    |   75.99   |  74.58  |  46.25  |  69.00   |
> | SliceGPT |  53.97  |   53.04    |   32.65   |  34.76  |  23.72  |  39.63   |
> | PruneNet |  59.08  |   52.09    |   35.21   |  34.89  |  25.43  |  41.46   |
>
> **Format: Sparsity and Effective Sparsity.** We thank the reviewer for the suggestion. We have added the explanation in Appendix A.2 of the updated paper.
>
> **References**
>
> [1] Yuan, Zhihang, Yuzhang Shang, Yue Song, Qiang Wu, Yan Yan, and Guangyu Sun. "Asvd: Activation-aware singular value decomposition for compressing large language models." arXiv preprint arXiv:2312.05821 (2023).
>
> [2] Li, Yuanzhi, Sébastien Bubeck, Ronen Eldan, Allie Del Giorno, Suriya Gunasekar, and Yin Tat Lee. "Textbooks are all you need ii: phi-1.5 technical report." arXiv preprint arXiv:2309.05463 (2023).

---

> > ### Comment · Reviewer_onBH · 2024-11-21
> > **Review for rebuttal**
> >
> > Thanks for the detailed rebuttal. The authors have addressed all my questions and I change my score accordingly. All the best with your submission.

---

> ### Author Response · Authors · 2024-11-22
> **Acknowleding Reviewer onBH's Comment**
>
> Dear reviewer onBH,
>
> We thank you for reassessing our paper and increasing the soundness score. We would greatly appreciate your guidance on any additional concerns you may have about improving the **overall rating of our paper**.
>
> We are eager to hear any additional recommendations for increasing the impact of our work.
>
> Thanks,

---

### Official Review · Reviewer_ooUi · 2024-11-01

**Soundness:** 3
**Presentation:** 1
**Contribution:** 3
**Rating:** 6
**Confidence:** 3

**Summary:**

This paper introduces PruneNet, a structured pruning technique that deploys a policy learning process to identify important parameters. The proposed method is calibration-free, and once completed, can be applied to different pruning ratios without repeated assmentments of weights and data. The proposed method can be applied to popular Llama-2-7B in just 15 minutes, which is efficient enough for LLM pruning. The author conducted experiments on Llama-2 7B and Phi-2, with the compression ratio from 20% to 30%. Compared to SliceGPT, the method archives better accuracy on 5 zero-shot tasks, keeping >80% performance of the original LLMs. Besides, even without fine-tuning, the method is still able to preserve a good average accuracy.

**Strengths:**

The idea of calibration-free pruning is interesting.  The policy learned on one compression ratio is scalable and can be directly transferred to higher or lower ratios. This is a useful feature if one would like to evaluate the performance of pruned models for a good trade-off between performance and efficiency. In addition, Extensive experiments were conducted to evaluate the effectiveness of the policy learning, with different finetuning settings and compression ratios, The proposed method is able to achieve better average accuracy (+3%~+4%) on zero-shot tasks.

**Weaknesses:**

1. It appears that most of the experiments in this paper were conducted without fine-tuning. It would be insightful to examine how the proposed method compares to other baselines when fine-tuning is enabled, as this could provide a more complete picture of its performance under optimal conditions.
2. The efficiency of policy learning has not been studied. Providing more analysis on the efficiency of policy learning would enhance the understanding of its practical impact.
3. Based on point 3, if we allocate the same computational budget across all baselines, such as assigning the training cost of policy learning to LoRA fine-tuning in other baselines, does the proposed method still achieve superior performance?
4. Including pseudocode for the entire pipeline would improve clarity and reproducibility.
5. On of my main concerns is the lack of clarity behind the motivation for each design choice. For example, Equation 3 is introduced without sufficient context or explanation. The KS Distance introduced in Equation 6 also reveals weak insights about its motivation. Why KS distance instead of other distribution distance? In addition, other paragraphs could also benefit from refinement. In particular, the background section, especially sections 3.2 and 3.2, is densely packed with technical details and formulas about SliceGPT and eigenvalues. However, the relevance to the proposed method remains unclear until the final paragraph, creating a disconnect that might hinder reader understanding.

**Questions:**

May I ask about the motivation behind Equation 3 (the importance score)? It appears that the equation is designed to project the parameter $W$ into a score $d$-dim vector. Could you clarify why this approach was chosen over other designs, such as directly optimizing a d-dimensional vector as the importance score? Additionally, will the learned matrix be shared across all linear weight layers?

---

> ### Author Response · Authors · 2024-11-19
> **Response to Reviewer ooUi Comments - Part 1**
>
> We thank the reviewer for the valuable feedback. We address the concerns raised. All the changes in the main manuscript are highlighted with blue color.
>
> **Comparison of PruneNet with other compression methods with RFT enabled.** We report the results with LLaMA-2-7B with RFT on Wikitext2 and Alpaca datasets in Table 1 (Table 11 of the updated paper) and Table 2 (Table 12 of the updated paper), respectively. With RFT on the Wikitext2 dataset, PruneNet achieves, on average, $>5$% accuracy than SliceGPT. However, SliceGPT outperforms PruneNet when fine-tuned on the Alpaca dataset, with an average margin of $3$%. As SliceGPT uses the same datasets for calibration and RFT, it typically has access to more instruction-tuning datasets than PruneNet, allowing it to do better when fine-tuned on the Alpaca dataset. However, it is worth noticing that the average standard deviation in the performance of SliceGPT after RFT is significantly high ($5.5$) compared to PruneNet ($1.5$). The low standard deviation highlights the robustness of PruneNet when fine-tuned on different datasets to recover information loss during compression.
>
> ### Table 1. Results with RFT on WikiText2 on Llama-2-7B
>
> | Compression Ratio |   Method   |  PIQA   | WinoGrande | HellaSwag |  ARC-e  |  ARC-c  |   Avg.   |
> |       -----       | ---------- | ------- |  -------   |  -------  | ------- | ------- | -------- |
> |        0%         |   Dense    |  79.11  |   69.06    |   75.99   |  74.58  |  46.25  |  69.00   |
> |        20%        |  SliceGPT  |  69.86  |   64.72    |   61.07   |  54.25  |  36.73  |  57.27   |
> |        20%        |  PruneNet  |  74.76  |   66.22    |   69.38   |  65.61  |  39.25  |  63.04   |
> |        25%        |  SliceGPT  |  69.26  |   64.96    |   58.65   |  52.36  |  35.75  |  56.20   |
> |        25%        |  PruneNet  |  74.37  |   66.46    |   65.71   |  60.82  |  36.60  |  60.79   |
> |        30%        |  SliceGPT  |  67.41  |   63.22    |   55.65   |  50.76  |  34.13  |  54.23   |
> |        30%        |  PruneNet  |  73.01  |   63.46    |   63.21   |  60.14  |  35.92  |  59.15   |
>
> ### Table 2. Results with RFT on Alpaca on Llama-2-7B
>
> | Compression Ratio |   Method   |  PIQA   | WinoGrande | HellaSwag |  ARC-e  |  ARC-c  |   Avg.   |
> |       -----       | ---------- | ------- |  -------   |  -------  | ------- | ------- | -------- |
> |        0%         |   Dense    |  79.11  |   69.06    |   75.99   |  74.58  |  46.25  |  69.00   |
> |        20%        |  SliceGPT  |  76.55  |   65.59    |   68.26   |  71.84  |  45.05  |  65.46   |
> |        20%        |  PruneNet  |  72.73  |   62.25    |   66.45   |  61.52  |  42.15  |  61.02   |
> |        25%        |  SliceGPT  |  75.79  |   63.22    |   65.12   |  68.22  |  42.83  |  63.04   |
> |        25%        |  PruneNet  |  75.79  |   62.35    |   65.48  |  60.94  |  39.16  |  60.74   |
> |        30%        |  SliceGPT  |  74.59  |   61.64    |   63.06   |  66.54  |  40.87  |  61.34   |
> |        30%        |  PruneNet  |  72.14  |   62.75    |   62.38   |  55.43  |  37.03  |  57.95   |

---

> ### Author Response · Authors · 2024-11-19
> **Response to Reviewer ooUi Comments - Part 2**
>
> **Efficiency of policy learning.** The primary motivation behind policy learning is to decouple the model pruning process from the model itself. Therefore, with an appropriate policy learner model, we can learn which parameters can be compressed within an arbitrary model component just by looking at the model parameters without posing any additional constraint on the component architecture. This allows us to flexibly use the same policy learner model for compressing different layers and components of an LLM.
>
> We highlight the results with random policy in Table 3 (Table 16 of the updated paper), where we select the pruned indices randomly for each model parameter. We refer to this as a 'random' selection. We observe an average $2$% drop with a random selection method, demonstrating the need for a learnable policy for effective model compression.
>
> ### Table 3. Effect of learnable policy on compressed Llama-2-7B model.
>
> | Compression Ratio |   Selection   |  PIQA   | WinoGrande | HellaSwag |  ARC-e  |  ARC-c  |   Avg.   |
> |       -----       | ---------- | ------- |  -------   |  -------  | ------- | ------- | -------- |
> |        20%        |   Policy-based   |  75.30  |   65.50    |   66.43   |  63.80  |  37.29  |  61.66   |
> |        20%        |   Random   |  72.36  |   63.14    |   61.18   |  60.31  |  36.52  |  58.70   |
> |        25%        |   Policy-based   |  72.09  |   62.43    |   62.33   |  60.14  |  36.18  |  58.63   |
> |        25%        |   Random   |  70.13  |   60.38    |   58.38   |  56.27  |  35.67  |  56.20   |
> |        30%        |   Policy-based   |  71.13  |   61.09    |   58.30   |  53.20  |  33.53  |  55.45   |
> |        30%        |   Random   |  73.13  |   59.35    |   55.15   |  49.83  |  31.66  |  53.90   |
>
> To further understand the effectiveness of the learned policy empirically, we perform experiments where we learn deterministic policy based on the parameter importance calculated in Equation 3 of the paper. In the deterministic policy, we chose only the topk parameters based on this computed importance metric, making the selection deterministic. Table 4 (Table 17 of the updated paper) highlights the results of LLaMA-2-7B with deterministic and stochastic policies (policy learned with PruneNet defined in Equation 5 of the paper). We observe that deterministic policy often underperforms the stochastic policy with an average margin of $2$%. The results highlight that parameter importance alone cannot determine which parameters to compress. Preserving the spectral structure between the compressed and uncompressed models is critical to ensure minimal performance drop post-compression.
>
> ### Table 4. Effect of stochastic policy on compressed Llama-2-7B model.
>
> | Compression Ratio |    Policy     |  PIQA   | WinoGrande | HellaSwag |  ARC-e  |  ARC-c  |   Avg.   |
> |       -----       |  ----------   | ------- |  -------   |  -------  | ------- | ------- | -------- |
> |        20%        |  Stochastic   |  75.30  |   65.50    |   66.43   |  63.80  |  37.29  |  61.66   |
> |        20%        | Deterministic |  72.91  |   61.64    |   61.05   |  56.69  |  36.52  |  57.76   |
> |        25%        |  Stochastic   |  72.09  |   62.43    |   62.33   |  60.14  |  36.18  |  58.63   |
> |        25%        | Deterministic |  69.97  |   59.27    |   59.39   |  56.69  |  33.53  |  55.77   |
> |        30%        |  Stochastic   |  71.13  |   61.09    |   58.30   |  53.20  |  33.53  |  55.45   |
> |        30%        | Deterministic |  69.64  |   58.25    |   54.45   |  54.97  |  31.23  |  53.71   |
>
> **Computational budget.** The policy learner model is typically $1000$ times smaller than the LLM to be compressed (number of learnable parameters $<7$M). As Section 1 of the paper highlights, most existing structured pruning methods learn the compressed model after being calibrated on some dataset. Therefore, the computational budget utilized by the policy learner model is far less than the existing baselines, making our compression method more scalable and efficient.
>
> **Pseudo code.** We thank the reviewer for the suggestion. We have added the pseudo-code of the proposed compression method in Algorithm 1 of the updated paper.

---

> ### Author Response · Authors · 2024-11-19
> **Response to Reviewer ooUi Comments - Part 3**
>
> **A detailed motivation behind design choices.** We perform an ablation study to understand the importance of different components of PruneNet.
>
> We conduct experiments with a random selection process, where the pruned parameter indices are chosen randomly. We highlight the results with random policy in the above Table 3 (Table 16 of the updated paper). We observe an average $2$% drop with a random selection method, justifying the need to learn which parameters to compress for more effective model compression.
>
> Above Table 4 (Table 17 of the updated paper) highlights the results with LLaMA-2-7B with deterministic and stochastic (policy learned with PruneNet in Equation 5 of the paper) policies.  In the deterministic policy, we chose only the topk parameters based on the importance metric defined in Equation 3. We observe that deterministic policy often underperforms the stochastic policy with an average margin of $4$%. The results highlight that parameter importance alone cannot determine which parameters to compress. Preserving the spectral structure between the compressed and uncompressed models is critical to ensure minimal performance drop post-compression.
>
> To further understand the effectiveness of PruneNet under different distance measures, we evaluate the LLaMA-2-7B model compressed using PruneNet with non-parametric Anderson–Darling measure of agreement. Table 5 (Table 18 of the updated paper) highlights the effectiveness of PruneNet with both Kolmogorov-Smirnov (highlighted as KS) and  Anderson–Darling (highlighted as AD) distance measures. Under both reward functions, we observe a similar performance of PruneNet for different compression ratios with the LLaMA-2-7B model. The results further emphasize the stability of our proposed compression method under different choice metrics.
>
> ### Table 5. Zero-shot performance of Llama-2-7B compressed with PruneNet with different reward functions.
>
> | Compression Ratio | Reward Function |  PIQA   | WinoGrande | HellaSwag |  ARC-e  |  ARC-c  |   Avg.   |
> |       -----       |   ----------    | ------- |  -------   |  -------  | ------- | ------- | -------- |
> |        20%        |       KS        |  75.30  |   65.50    |   66.43   |  63.80  |  37.20  |  61.66   |
> |        20%        |       AD        |  73.01  |   63.30    |   65.70   |  60.40  |  37.46  |  59.97   |
> |        25%        |       KS        |  72.09  |   62.43    |   62.33   |  60.14  |  36.18  |  58.63   |
> |        25%        |       AD        |  73.88  |   61.17    |   63.98   |  61.62  |  35.84  |  59.30   |
> |        30%        |       KS        |  71.13  |   61.09    |   58.30   |  53.20  |  33.53  |  55.45   |
> |        30%        |       AD        |  72.13  |   61.88    |   60.18   |  58.00  |  33.62  |  57.16   |
>
> Table 6 (Table 19 of the updated paper) highlights the zero-shot performance of the LLaMA-2-7B model compressed with PruneNet with policy learned on different FFN matrices. In most cases, we observe marginal differences in the result ($< 1$%) when the policy is learned with FFN2 instead of FFN1. The observations emphasize that PruneNet is invariant to the choice of parameter used for learning the policy.
>
> ### Table 6. A comparison of FFN1 vs FFN2 layers within the PruneNet framework for Llama-2-7B.
>
> | Compression Ratio |   Layer    |  PIQA   | WinoGrande | HellaSwag |  ARC-e  |  ARC-c  |   Avg.   |
> |       -----       | ---------- | ------- |  -------   |  -------  | ------- | ------- | -------- |
> |        20%        |    FFN1    |  75.30  |   65.50    |   66.43   |  63.80  |  37.29  |  61.66   |
> |        20%        |    FFN2    |  74.81  |   66.93    |   67.38   |  61.24  |  36.86  |  61.44   |
> |        25%        |    FFN1    |  72.09  |   62.43    |   62.33   |  60.14  |  36.18  |  58.63   |
> |        25%        |    FFN2    |  70.13  |   57.30    |   59.98   |  55.51  |  34.22  |  55.43   |
> |        30%        |    FFN1    |  71.11  |   61.09    |   58.30   |  53.20  |  33.53  |  55.45   |
> |        30%        |    FFN2    |  72.20  |   61.56    |   60.01   |  54.12  |  33.70  |  56.32   |

---

> ### Author Response · Authors · 2024-11-21
> **Response to Reviewer ooUi Comments - Part 3 (contd.)**
>
> **Response to Question.** The key motivation behind the design of PruneNet is to have a predictor network which can compute the importance scores of the rows of *any* weight matrix; this allows us to reuse the same learned predictor network to prune the FFN weight matrix of *any* layer in the model, which turns out to be extremely efficient compared to other SOTA pruning methods.
>
> Some clarification about equation (3) of main text is in order. Note that the output of equation (3) is an $n$-dimensional vector (and not a $d$-dimensional vector), where $n$ is the number of rows in the weight matrix $W$. In essence, the predictor network is computing importance scores for each row of a weight matrix in two steps: in the first part of equation (3), the network computes the *relative importance* of rows amongst themselves. This is needed since the importance of a row is potentially correlated with the importance of other rows of the weight matrix. The second part of equation (3), particularly the matrix multiplication within $\sigma$, simply projects the $n\times n$ matrix of relative importance scores into an output vector of importance scores of each row. In contrast, as suggested by the reviewer, directly optimizing a vector of importance scores conflicts with our design philosophy of **reusability** of the learned predictor network.
>
> Regarding the use of a distributional distance: since pruning reduces the cardinality of the spectrum, the use of traditional pointwise distance measures (like Euclidean/Cosine/Frobenius norm) are not applicable for comparing the spectrum structure of uncompressed and compressed models. Due to this, we resort to distributional distance measures to capture the shift in the spectrum of model; a specific structural argument is given in Corollary 3.3. As mentioned above, we also evaluated the effectiveness of PruneNet under two different distance measures, namely the KS distance and the non-parametric Anderson-Darling measure of agreement.
>
> We also explained above that directly optimizing the parameter importance often leads to poorer performance than the learned compression policy. We reuse the same policy learner for all the layers within the LLM. The policy learner is agnostic to the model architecture and learns a mapping between the compressed and uncompressed model parameters. The simplicity of the policy learner allows us to reuse the same model to compress an LLM at any given compression ratio, offering greater flexibility and efficiency.

---

> > ### Author Response · Authors · 2024-11-22
> > **A gentle reminder to check our responses**
> >
> > Dear reviewer ooUi,
> >
> > We have addressed all the concerns you raised with additional empirical evidence. We request you to kindly review our responses and let us know if you have further questions.
> >
> > We will make our best effort to clarify your doubts and concerns.
> >
> > Thanks,

---

> > > ### Comment · Reviewer_ooUi · 2024-11-22
> > >
> > > Thanks for the detailed response. Will update the score to 6. Good luck!

---

> ### Author Response · Authors · 2024-11-22
> **Acknowledging Reviewer ooUi's Comment**
>
> Dear reviewer ooUi,
>
> We thank you for your effort in reassessing our work and increasing the overall rating. We are committed to addressing all your concerns into the paper.
>
> Thanks,

---

### Official Review · Reviewer_bk7b · 2024-11-02

**Soundness:** 2
**Presentation:** 3
**Contribution:** 2
**Rating:** 6
**Confidence:** 3

**Summary:**

This paper proposes a FFN-layer pruning algorithm for large language models that does not require a calibration dataset. The idea is to prune the weight matrices so that the spectrum of the matrices before and after pruning remains similar in the Kolmogorov-Smirnoff distance. To hasten the pruning procedure, one trains a policy that predicts the importance scores of the first layer of FFN. The empirical results show that the proposed method outperforms SliceGPT, in terms of prediction quality and the actual speedup.

**Strengths:**

- The central research question of training a policy to prune LLMs is quite intriguing, with some practical importance. Although the paper mainly claims the benefit on the data side, I personally find that the method might have more future uses on low-memory or compute scenarios where it is very difficult to run usual pruning algorithms that require much gradient-based computations.

- The empirical performance seems to be reasonably good.

- The paper asks many interesting questions in section 5.2, including the transferability and the layerwise compression ratio.

**Weaknesses:**

- One nitpick is that the comparison against SliceGPT, reported on Table 2, can be quite misleading. The figures seems to be the one for SliceGPT with WikiText2 calibration data, which is much worse than the SliceGPT with Alpaca calibration data. In fact, with Alpaca, I believe that the SliceGPT works much better than the proposed PruneNet. Of course, this is somewhat expected because it utilizes more data than PruneNet; making a comparison with both WikiText and Alpaca case does not diminish the usefulness of the proposed method, as they assume different amount of usable resources. Thus I recommend including both SliceGPTs, for a more comprehensive comparison.

- Another weakness is that the proposed method may not work better than the methods that use calibration data, as assuming no access to the calibration data also makes it impossible to use RFT. It seems like Table 3, ironically, can also be interpreted as a limitation of the proposed method, in a sense that the model pruned with PruneNet cannot recover much with RFT, while LLMs pruned with other methods can recover much with fine-tuning.

- Also, the motivation is quite unclear to me. At least for the text data, I am not sure why it is practical to assume no access to the calibration data. We already have quite abundant text data (crawled from web), even as our benchmark. Why should we suddenly assume that those data are not available? Can you give any further justifications?

- The idea of spectrum matching is also not justified well. First, I do not see how corollary 3.3 can be used to predict that the singular values of the matrix become more right-skewed. Doesn't the corollary simply states that the spectrum will be about subsampling, rather than predicting anything about the skewness? Also, the fact that there can be some spectrum shift does not fully mean that it will be an effective criterion for deciding how to prune. Perhaps more comparison with other criterion (e.g., minimizing Frobenius norm distortion) may help us understand whether the spectrum matching part is indeed an essential and nontrivial component.

**Questions:**

Discussed in 'weaknesses'

---

> ### Author Response · Authors · 2024-11-19
> **Response to Reviewer bk7b Comments - Part 1**
>
> We thank the reviewer for the valuable feedback. We address the concerns raised. All the changes in the main manuscript are highlighted with blue color.
>
> **Comparison with SliceGPT with Alpaca calibration data.** Table 1 (Table 10 of the updated paper) contains a comparison of the performance of the LLaMA-2-7B and Phi-2 pruned with SliceGPT using the Alpaca calibration dataset (as opposed to the WikiText2 dataset used in the main text of our paper). While LLaMA-2-7B pruned with SliceGPT exhibits better average performance for all compression ratios, Phi-2 exhibits an opposite trend, wherein PruneNet beats SliceGPT by a consistent margin of at least $2$% for all compression ratios. This trend reinforces a fundamental limitation of calibration data-based pruning techniques, namely their sensitivity to the choice and quality of the calibration dataset and the choice of model architectures. In conjunction with this, it should also be noted that the Alpaca dataset is an instruction-tuning dataset (as opposed to WikiText2). Therefore, it is not unreasonable for the performance of pruned models calibrated with the Alpaca dataset on generative tasks to be better than that of a calibration-free technique like PruneNet.
>
> ### Table 1. Results without RFT, where SliceGPT uses Alpaca as calibration data
>
> |    Model     | Compression Ratio |   Method   |  PIQA   | WinoGrande | HellaSwag |  ARC-e  |  ARC-c  |   Avg.   |
> | ------------ |       -----       | ---------- | ------- |  -------   |  -------  | ------- | ------- | -------- |
> |  Llama-2-7B  |        0%         |   Dense    |  79.11  |   69.06    |   75.99   |  74.58  |  46.25  |  69.00   |
> |  Llama-2-7B  |        20%        |  SliceGPT  |  76.5   |   65.51    |   65.2    |  69.99  |  41.21  |  63.68   |
> |  Llama-2-7B  |        20%        |  PruneNet  |  75.3   |   65.51    |   66.43   |  63.8   |  37.29  |  61.67   |
> |  Llama-2-7B  |        25%        |  SliceGPT  |  74.21  |   64.01    |   60.55   |  66.88  |  38.91  |  60.91   |
> |  Llama-2-7B  |        25%        |  PruneNet  |  72.09  |   62.43    |   62.33   |  60.14  |  36.18  |  58.63   |
> |  Llama-2-7B  |        30%        |  SliceGPT  |  72.25  |   59.83    |   55.86   |  63.93  |  37.8   |  57.93   |
> |  Llama-2-7B  |        30%        |  PruneNet  |  71.11  |   61.09    |   58.3    |  53.2   |  33.53  |  55.45   |
> |    Phi-2     |        0%         |   Dense    |  79.11  |   75.77    |   73.83   |  78.32  |  54.18  |  72.24   |
> |    Phi-2     |        20%        |  SliceGPT  |  76.17  |   68.75    |   61.95   |  72.18  |  45.48  |  64.90   |
> |    Phi-2     |        20%        |  PruneNet  |  74.37  |   70.80    |   65.53   |  74.71  |  47.53  |  66.59   |
> |    Phi-2     |        25%        |  SliceGPT  |  75.68  |   64.88    |   58.19   |  70.41  |  43.43  |  62.52   |
> |    Phi-2     |        25%        |  PruneNet  |  74.37  |   68.98    |   62.18   |  70.54  |  44.45  |  64.10   |
> |    Phi-2     |        30%        |  SliceGPT  |  74.05  |   62.12    |   53.31   |  67.26  |  39.42  |  59.23   |
> |    Phi-2     |        30%        |  PruneNet  |  72.80  |   67.48    |   56.80   |  67.55  |  40.61  |  61.05   |
>
> **Motivation for calibration-free model compression techniques.** While it is true that there is an abundance of calibration data for textual modalities, it is not the *availability* of calibration data alone that determines the quality of a pruned model. A detailed study by William et al., 2024 reveals a large degree of sensitivity in the performance on downstream tasks of pruned models based on the selected calibration data and model architectures, which questions the usability of calibration-based pruning methods in data-private settings; this has made the problem of sampling high-quality calibration datasets an active area of model pruning research. Another major limitation of calibration-based pruning techniques is that of *reusability*, wherein pruning multiple models necessitates running the forward/backward passes of pruned models on the calibration datasets, making the overall process highly inefficient. In contrast, PruneNet aims to be the *first* pruning technique to alleviate these difficulties and can also run right out of the box in low-resource/compute settings. Finally, PruneNet can be used in data modalities other than text, where sampling calibration data might be costly.

---

> ### Author Response · Authors · 2024-11-19
> **Response to Reviewer bk7b Comments - Part 2**
>
> **Justification of spectrum matching and comparison with other reward criterion.** It is important to understand that pruning reduces the cardinality of the spectrum. Therefore, the traditional pointwise distance measures (like Euclidean/Cosine/Frobenius norm) are not applicable for comparing the spectrum structure of uncompressed and compressed models. Therefore, we resort to probabilistic distance measures that can capture the shift in the empirical distribution of spectrums of the compressed model. Corollary 3.3 suggests that compression reduces the spectrum range, and Figure 1 highlights that with more compression, the spectrum becomes more right-skewed. To further understand the effectiveness of PruneNet under different distance measures, we evaluate the LLaMA-2-7B model compressed using PruneNet with non-parametric Anderson–Darling measure of agreement.
>
> Table 2 (Table 18 of the updated paper) highlights the effectiveness of PruneNet with both Kolmogorov-Smirnov (highlighted as KS) and  Anderson–Darling (highlighted as AD) distance measures. Under both reward functions, we observe a similar performance of PruneNet for different compression ratios with the LLaMA-2-7B model. The results further emphasize the stability of our proposed compression method under different choice metrics.
>
> ### Table 2. Zero-shot performance of Llama-2-7B compressed with PruneNet with different reward functions.
>
> | Compression Ratio | Reward Function |  PIQA   | WinoGrande | HellaSwag |  ARC-e  |  ARC-c  |   Avg.   |
> |       -----       |   ----------    | ------- |  -------   |  -------  | ------- | ------- | -------- |
> |        20%        |       KS        |  75.30  |   65.50    |   66.43   |  63.80  |  37.20  |  61.66   |
> |        20%        |       AD        |  73.01  |   63.30    |   65.70   |  60.40  |  37.46  |  59.97   |
> |        25%        |       KS        |  72.09  |   62.43    |   62.33   |  60.14  |  36.18  |  58.63   |
> |        25%        |       AD        |  73.88  |   61.17    |   63.98   |  61.62  |  35.84  |  59.30   |
> |        30%        |       KS        |  71.13  |   61.09    |   58.30   |  53.20  |  33.53  |  55.45   |
> |        30%        |       AD        |  72.13  |   61.88    |   60.18   |  58.00  |  33.62  |  57.16   |
>
> **References**
>
> [1] Williams, Miles, and Nikolaos Aletras. "On the impact of calibration data in post-training quantization and pruning." In Proceedings of the 62nd Annual Meeting of the Association for Computational Linguistics (Volume 1: Long Papers), pp. 10100-10118. 2024.

---

> > ### Author Response · Authors · 2024-11-22
> > **A gentle reminder to check our responses**
> >
> > Dear reviewer bk7b,
> >
> > We have addressed all the concerns you raised with additional empirical evidence. We request you to kindly review our responses and let us know if you have further questions.
> >
> > We will make our best effort to clarify your doubts and concerns.
> >
> > Thanks,

---

> > ### Author Response · Authors · 2024-11-23
> > **Please read our responses**
> >
> > Dear reviewer bk7b,
> >
> > The discussion period ends very soon. This is our sincere request to read our responses. We tried our best to address all your comments with additional experimental results. Kindly let us know if you have any further questions. If our responses address your concerns, kindly consider reassessing our paper.
> >
> > We look forward to your response.
> >
> > Thanks

---

> > ### Author Response · Authors · 2024-11-24
> > **Please check our response**
> >
> > Dear reviewer bk7b,
> >
> > The discussion period is ending very soon. We are yet to receive any feedback from you regarding our responses. We tried our best to address all your comments with additional experimental results. This is our sincere request to you to kindly check our responses and consider reassessing our paper.
> >
> > We look forward to your response.
> >
> > Thanks

---

> > > ### Comment · Reviewer_bk7b · 2024-11-25
> > >
> > > Thank you for the detailed response. I have just read through your comments, and came up with several follow-up questions.
> > >
> > > **On Alpaca.** Thank you for sharing an interesting result. The outperformance of the proposed method on Phi-2 is in fact quite interesting. Yet, I do observe that there is some discrepancy between the performance of SliceGPT on Phi-2 you reported and the ones in the Table 8 of the original paper. Why is that?
> > >
> > > Also, there is nothing wrong if data-free method to slightly underperform a method that utilizes data extensively; it does not undermine quality of the paper (or make me appreciate this paper less). My point is that a comprehensive and informative comparison is always better; thus not including them in the paper is indeed something that hurts the quality of the paper.
> > >
> > > **On RFT.** Would there be any thoughts on this point?
> > >
> > > **On calibration data.** Thank you for the pointer to Williams and Aletras (2024). I do agree to the point on the reusability. Regarding the sensitivity, it is not clear to me why this should be the good argument for not using the calibration data at all; wouldn't a careful choice of calibration data (based on many experiments) be the obvious choice? Also, regarding the data modalities other than text, it would be great if authors could give a nice example.
> > >
> > > **On spectrum matching.** I see. It is still questionable to me how important the theoretical results are to the actual design decisions here, but is nevertheless useful to have one.

---

> > > > ### Author Response · Authors · 2024-11-25
> > > > **Response to Reviewer bk7b Comments - Part I**
> > > >
> > > > We thank reviewer bk7b for the comments. We address the concerns below -
> > > >
> > > > **Discrepancy in the tables.** The discrepancy is in SliceGPT's original paper (Askboos et al. 2024), where they computed the average score incorrectly for the 30% sparsity ratio. We report the correct number in the above response.
> > > >
> > > > We have updated the Alpaca results in the revised paper.
> > > >
> > > > **RFT.** We argue that, unlike other compression methods like SliceGPT, RFT is optional for recovering models compressed with PruneNet. As PruneNet preserves the internal knowledge of an LLM through spectral matching, PruneNet is less influenced by RFT. Nevertheless, results in Table 1 show that the post-RFT performance of PruneNet is still better than SliceGPT's.
> > > >
> > > > ### Table 1. Results with RFT on WikiText2 on Llama-2-7B
> > > >
> > > > | Compression Ratio |   Method   |  PIQA   | WinoGrande | HellaSwag |  ARC-e  |  ARC-c  |   Avg.   |
> > > > |       -----       | ---------- | ------- |  -------   |  -------  | ------- | ------- | -------- |
> > > > |        0%         |   Dense    |  79.11  |   69.06    |   75.99   |  74.58  |  46.25  |  69.00   |
> > > > |        20%        |  SliceGPT  |  69.86  |   64.72    |   61.07   |  54.25  |  36.73  |  57.27   |
> > > > |        20%        |  PruneNet  |  74.76  |   66.22    |   69.38   |  65.61  |  39.25  |  63.04   |
> > > > |        25%        |  SliceGPT  |  69.26  |   64.96    |   58.65   |  52.36  |  35.75  |  56.20   |
> > > > |        25%        |  PruneNet  |  74.37  |   66.46    |   65.71   |  60.82  |  36.60  |  60.79   |
> > > > |        30%        |  SliceGPT  |  67.41  |   63.22    |   55.65   |  50.76  |  34.13  |  54.23   |
> > > > |        30%        |  PruneNet  |  73.01  |   63.46    |   63.21   |  60.14  |  35.92  |  59.15   |
> > > >
> > > > **On calibration data.** We emphasize that a careful selection of calibration datasets, which may require extensive experimentation, is a significant limitation of existing compression methods and may restrict their applicability in many applications. For instance, consider a scenario where an LLM has been trained on private data for domain-specific applications such as healthcare. Finding good-quality calibration datasets might be monetarily and legally demanding in such a situation. Moreover, evaluating the post-compression performance of LLMs to choose the correct calibration could also be infeasible in those scenarios.
> > > >
> > > > Another point that motivated us to pursue data-free pruning methods was that calibration-based pruning methods perform computation in the original model space, which requires loading the calibration data in memory and running it through the model. For instance - SliceGPT performs rotation operations which are performed on CPUs (reference - https://github.com/microsoft/TransformerCompression/blob/main/src/slicegpt/rotate.py) to prevent out-of-GPU-memory errors. Due to these computational challenges, SliceGPT requires a significantly longer time to compress LLMs than PruneNet (highlighted in Section 5.3 of the paper).
> > > >
> > > > On the other hand, RFT is often performed with parameter-efficient fine-tuning strategies such as LoRA. Therefore, fine-tuning a compressed LLM is computationally more viable than calibration. PruneNet outperforms SliceGPT both with and without RFT, demonstrating its effectiveness without introducing significant computational expense.
> > > >
> > > > **Example of multi-modal calibration.** Compressing a multi-modal LLM like Video-LLaVA-7B (Lin et al., 2023) using traditional methods might require good-quality video calibration data. PruneNet offers greater flexibility by omitting the need for calibration. Therefore, PruneNet can easily compress this model without spending any extra effort collecting a video dataset.

---

> > > > ### Author Response · Authors · 2024-12-02
> > > > **Possibly last reminder to check our comments**
> > > >
> > > > Dear reviewer bk7b,
> > > >
> > > > It is unfortunate that even after so many reminders we were not able to receive any further response from your end. We tried our best to provide answers to all the queries you raised. This is our final request to you to check our previous comments. We are ready to address other concerns if you have any.
> > > >
> > > > Thanks,

---

> > > > > ### Comment · Reviewer_bk7b · 2024-12-03
> > > > >
> > > > > Dear authors,
> > > > >
> > > > > Sorry for the late comeback.
> > > > >
> > > > > I appreciate much discussion regarding the availability of validation dataset. I strongly recommend including the discussion to the manuscript. Although I do have some unclear points regarding the spectrum matching, I am now convinced that this paper makes meaningful progress. Have raised my evaluation accordingly.
> > > > >
> > > > > Best regards,
> > > > > Authors.

---

> > > > > > ### Author Response · Authors · 2024-12-03
> > > > > > **Thank you**
> > > > > >
> > > > > > Dear Reviewer bk7b,
> > > > > >
> > > > > > Thank you very much for your response and for raising the score. We are committed to incorporate all your suggestions into the manuscript.
> > > > > >
> > > > > > Thanks

---

> ### Author Response · Authors · 2024-11-25
> **Response to Reviewer bk7b Comments - Part II**
>
> **On spectrum matching.** In Figure 1 of our paper, we highlighted how slicing impacts the spectrum structure of an LLM. In Corollary 3.3, we formalized this fact using the Poincare separation theorem. To understand the importance of using a learned policy instead of a static method such as random slicing of a weight matrix, we further carried out the following experiment: we randomly pick indices to slice off for each layer and report the standard deviation and mean of the obtained rewards. In Table 2, we observe a high standard deviation of rewards for smaller compression ratios, thereby requiring a mechanism that **learns** the indices to be pruned off.
>
> | Sparsity | Mean Reward (Layer 1) | STD Reward (Layer 1) | Mean Reward (Layer 32) | STD Reward (Layer 32) |
> | -------- | ------------ | -----------  | ------------- | ------------ |
> | 5%       | 30.9         | 1.04         | 31.22         | 0.31         |
> | 10%      | 16.2         | 0.39         | 15.6          | 0.18         |
> | 20%      | 8.2          | 0.07         | 7.7           | 0.03         |
> | 25%      | 6.6          | 0.06         | 6             | 0.04         |
> | 40%      | 4.2          | 0.05         | 3.6           | 0.01         |
> | 55%      | 3.1          | 0.02         | 2.5           | 0.01         |
> | 70%      | 2.3          | 0.02          | 1.8           | 0            |
>
> Therefore, Corollary 3.3 and the above experiment suggest developing a learnable strategy to minimize the difference between uncompressed and compressed model spectrums for retaining information in the post-compressed LLM.
>
> **References**
>
> [1] Ashkboos, Saleh, Maximilian L. Croci, Marcelo Gennari do Nascimento, Torsten Hoefler, and James Hensman. "Slicegpt: Compress large language models by deleting rows and columns." arXiv preprint arXiv:2401.15024 (2024).
>
> [2] Lin, Bin, Yang Ye, Bin Zhu, Jiaxi Cui, Munan Ning, Peng Jin, and Li Yuan. "Video-llava: Learning united visual representation by alignment before projection." arXiv preprint arXiv:2311.10122 (2023).

---

> > ### Author Response · Authors · 2024-11-25
> > **Please check our responses to your followup questions**
> >
> > Dear Reviewer bk7b,
> >
> > Thank you very much for your followup questions. We have addressed your recent concerns with new sets of experiments. Since the discussion is ending tomorrow, we request you to have a look at our recent responses and let me know if you have further concerns. If you feel our responses address your questions, please consider reassessing our paper.
> >
> > Looking forward to more followup discussion.
> >
> > Thanks

---

> ### Author Response · Authors · 2024-11-27
> **Sincere request to check our recent responses**
>
> Dear reviewer bk7b,
>
> We have addressed your recent concerns with new sets of experiments. Since the discussion period is ending soon, we request you to have a look at our recent responses and let us know if you have further concerns. Your feedback is valuable for us to address the potential weaknesses and improving the quality of our work.
>
> If you feel our responses address your questions, please consider reassessing our paper.
>
> Thanks

---

> > ### Author Response · Authors · 2024-11-27
> > **Could you please check our response?**
> >
> > Dear reviewer bk7b,
> >
> > We have addressed your recent concerns with new sets of experiments. Could you please check our responses? If you feel our responses address your questions, please consider reassessing our paper. We are ready to address other concerns if any?
> >
> > Thanks

---

> > > ### Author Response · Authors · 2024-11-28
> > > **Subsequent reminder to read our responses**
> > >
> > > Dear bk7b,
> > >
> > > This is another reminder to check our responses to your recent comments. We have meticulously addressed all your new concerns. We sincerely request you check our responses.
> > >
> > > Thanks

---

> ### Author Response · Authors · 2024-11-30
> **Requesing again to check our responses**
>
> Dear reviewer bk7b,
>
> We have addressed your recent concerns with new sets of experiments. We sincerely request you check our responses. If you feel our responses address your questions, please consider reassessing our paper. We are ready to address other concerns if any.
>
> Thanks

---

> > ### Author Response · Authors · 2024-12-01
> > **Yet another request to check our responses**
> >
> > Dear reviewer bk7b,
> >
> > We only have a day left till the end of the discussion period. We earnestly request you to check our previous responses and comment back if you have any follow up question. We have put significant effort into conducting additional experiments and responded to your queries. Your acknowledgement is very important for us to present our work to the research community and give it the credit it deserves.
> >
> > Thanks,

---

### Official Review · Reviewer_ZGwT · 2024-11-03

**Soundness:** 3
**Presentation:** 3
**Contribution:** 3
**Rating:** 6
**Confidence:** 3

**Summary:**

This paper introduces PruneNet, a policy-based pruning method designed to learn the removal of redundant parameters without the need for calibration data. PruneNet’s policy is trained based on intrinsic model properties to minimize information loss, quantified using the KS distance between the singular value distributions of the uncompressed and compressed matrices. To make policy learning differentiable, the paper employs the reparameterization trick, introducing a random variable into the sampling process and training the policy module by evaluating various pruning outcomes. The proposed method is primarily evaluated on Llama-2 7B and Phi-2, with comparisons to SliceGPT, a technique that directly compresses parameter matrices by training on datasets. Results indicate that PruneNet achieves superior performance on several popular benchmarks.

**Strengths:**

* One of the key strengths of this paper is a data-independent importance scorer, derived through a policy-learning process. The policy focuses mainly on individual linear layers, making the method theoretically efficient. This efficiency is demonstrated by its successful application to Llama-7B, where pruning requires only 15 minutes, illustrating its practicality for large-scale models.
* The authors conducted extensive experiments, exploring factors such as compression ratio, RFT dataset, and computational costs.
* The policy learner can be applied across different compression ratios, which offers practical value for determining optimal pruning ratios in large language models (LLMs). This flexibility could be particularly advantageous for users looking to fine-tune pruning ratios without retraining the policy for each setting.

**Weaknesses:**

* The pruning approach is focused on single layers, potentially overlooking structured interdependencies with subsequent layers. In structured pruning, parameters in one layer impact those in the next; by not accounting for this, the proposed method risks removing essential parameters from interconnected layers. This oversight could inadvertently affect model performance, as it might miss opportunities for more coordinated pruning across layers.
* Although the method is designed to optimize the KS distance before and after pruning, it lacks a detailed analysis of its policy-learning approach. A simple random sampling approach might also work: sampling multiple randomly pruned matrices and selecting the one with the lowest KS distance might be able to provide good results. Including more ablation studies to compare these methods would enrich the analysis. Additionally, further discussion on the design and benefits of the policy approach would add valuable insights.
* While the method shows promise on Llama-7B, its efficacy on larger models, such as Llama 13B, remains untested. An evaluation on larger LLMs would help establish its robustness and scalability, further highlighting its applicability to a broader range of large models.

**Questions:**

Please see weaknesses

---

> ### Author Response · Authors · 2024-11-19
> **Response to Reviewer ZGwT Comments - Part 1**
>
> We thank the reviewer for the valuable feedback. We address the concerns raised. All the changes in the main manuscript are highlighted with blue color.
>
> **Structural interdependencies with subsequent layers.** We argue that the interdependencies between subsequent layers are already captured during the pre-training phase of the LLM. Therefore, the spectral structure of the feedforward layers already preserves the information propagation between the subsequent layers. By preserving the spectral structure after compression, PruneNet preserves the information that needs to be maintained at later layers, therefore omitting additional effort to preserve the information flow between interconnected layers.
>
> **A comparison with simple random sampling.** We highlight the results with random policy in Table 1 (Table 16 of the updated paper), where we select the pruned indices randomly for each model parameter. We refer to this as a 'random' selection. We observe an average $2$% drop with a random selection method, demonstrating the need for a learnable policy for effective model compression.
>
> ### Table 1. Effect of learnable policy on compressed Llama-2-7B model.
>
> | Compression Ratio |   Selection   |  PIQA   | WinoGrande | HellaSwag |  ARC-e  |  ARC-c  |   Avg.   |
> |       -----       | ---------- | ------- |  -------   |  -------  | ------- | ------- | -------- |
> |        20%        |   Policy-based   |  75.30  |   65.50    |   66.43   |  63.80  |  37.29  |  61.66   |
> |        20%        |   Random   |  72.36  |   63.14    |   61.18   |  60.31  |  36.52  |  58.70   |
> |        25%        |   Policy-based   |  72.09  |   62.43    |   62.33   |  60.14  |  36.18  |  58.63   |
> |        25%        |   Random   |  70.13  |   60.38    |   58.38   |  56.27  |  35.67  |  56.20   |
> |        30%        |   Policy-based   |  71.13  |   61.09    |   58.30   |  53.20  |  33.53  |  55.45   |
> |        30%        |   Random   |  73.13  |   59.35    |   55.15   |  49.83  |  31.66  |  53.90   |
>
> Moreover, Table 2 (Table 17 of the updated paper) highlights the results with LLaMA-2-7B with deterministic and stochastic (policy learned with PruneNet) policies. In the deterministic policy, we chose only the topk (k depends on the compression ratio) parameters based on the importance metric defined in Equation 3. We observe that deterministic policy often underperforms the stochastic policy with an average margin of $4$%. The results highlight that learnable parameters alone are insufficient to determine which ones to compress. Preserving the spectral structure between the compressed and uncompressed models is also critical to ensure minimal performance drop post-compression.\newline
>
> ### Table 2. Effect of stochastic policy on compressed Llama-2-7B model.
>
> | Compression Ratio |    Policy     |  PIQA   | WinoGrande | HellaSwag |  ARC-e  |  ARC-c  |   Avg.   |
> |       -----       |  ----------   | ------- |  -------   |  -------  | ------- | ------- | -------- |
> |        20%        |  Stochastic   |  75.30  |   65.50    |   66.43   |  63.80  |  37.29  |  61.66   |
> |        20%        | Deterministic |  72.91  |   61.64    |   61.05   |  56.69  |  36.52  |  57.76   |
> |        25%        |  Stochastic   |  72.09  |   62.43    |   62.33   |  60.14  |  36.18  |  58.63   |
> |        25%        | Deterministic |  69.97  |   59.27    |   59.39   |  56.69  |  33.53  |  55.77   |
> |        30%        |  Stochastic   |  71.13  |   61.09    |   58.30   |  53.20  |  33.53  |  55.45   |
> |        30%        | Deterministic |  69.64  |   58.25    |   54.45   |  54.97  |  31.23  |  53.71   |

---

> ### Author Response · Authors · 2024-11-19
> **Response to Reviewer ZGwT Comments - Part 2**
>
> **Efficacy of PruneNet on larger models.** Table 3 (Table 14 in the updated paper) highlights the results with PruneNet and SliceGPT for different compression ratios for the LLaMA-2-13B model. The performance drops drastically for larger models like LLaMA-13B at a high compression ratio. However, the results in Table 4 (Table 15 in the updated paper) highlight that after recovery fine-tuning, the compressed models regain the performance quickly and can preserve up to $84$% of the original uncompressed model performance, even at a high compression rate of $30$%. PruneNet also outperforms SliceGPT with a margin of $2$%, showcasing a similar trend as the smaller LLMs used in our study.
>
> ### Table 3. Llama-2-13B results without RFT.
>
> | Compression Ratio |   Method   |  PIQA   | WinoGrande | HellaSwag |  ARC-e  |  ARC-c  |   Avg.   |
> |       -----       | ---------- | ------- |  -------   |  -------  | ------- | ------- | -------- |
> |        0%         |   Dense    |  80.47  |   72.22    |   79.39   |  77.48  |  49.23  |  71.76   |
> |        20%        |  SliceGPT  |  71.87  |   69.38    |   63.04   |  69.87  |  43.09  |  63.45   |
> |        20%        |  PruneNet  |  77.15  |   66.38    |   72.90   |  70.50  |  41.81  |  65.75   |
> |        25%        |  SliceGPT  |  68.55  |   67.48    |   58.10   |  62.50  |  37.88  |  58.90   |
> |        25%        |  PruneNet  |  70.89  |   62.43    |   58.67   |  58.63  |  34.04  |  56.93   |
> |        30%        |  SliceGPT  |  66.10  |   65.11    |   52.69   |  56.82  |  35.07  |  55.16   |
> |        30%        |  PruneNet  |  61.92  |   56.99    |   35.65   |  46.34  |  28.33  |  45.87   |
>
> ### Table 4. Llama-2-13B results with RFT on WikiText2.
>
> | Compression Ratio |   Method   |  PIQA   | WinoGrande | HellaSwag |  ARC-e  |  ARC-c  |   Avg.   |
> |       -----       | ---------- | ------- |  -------   |  -------  | ------- | ------- | -------- |
> |        0%         |   Dense    |  80.47  |   72.22    |   79.39   |  77.48  |  49.23  |  71.76   |
> |        20%        |  SliceGPT  |  74.10  |   68.51    |   66.94   |  70.54  |  43.77  |  64.77   |
> |        20%        |  PruneNet  |  76.22  |   68.43    |   70.72   |  66.88  |  42.83  |  65.02   |
> |        25%        |  SliceGPT  |  71.27  |   68.98    |   64.12   |  63.76  |  40.87  |  61.80   |
> |        25%        |  PruneNet  |  76.93  |   64.80    |   70.44   |  66.96  |  40.36  |  63.90   |
> |        30%        |  SliceGPT  |  69.64  |   66.85    |   59.93   |  59.55  |  38.65  |  58.92   |
> |        30%        |  PruneNet  |  73.45  |   65.59    |   64.50   |  60.73  |  38.57  |  60.57   |

---

> > ### Author Response · Authors · 2024-11-22
> > **A gentle reminder to check our responses**
> >
> > Dear reviewer ZGwT,
> >
> > We have addressed all the concerns you raised with additional empirical evidence. We request you to kindly review our responses and let us know if you have further questions.
> >
> > We will make our best effort to clarify your doubts and concerns.
> >
> > Thanks,

---

> > ### Author Response · Authors · 2024-11-23
> > **Another reminder to read our response**
> >
> > Dear reviewer ZGwT,
> >
> > The discussion period ends very soon. It is our sincere request to read our responses. We tried our best to address all your comments with additional experimental results. Kindly let us know if you have any further questions. If our responses address your concerns, kindly consider reassessing our paper.
> >
> > We look forward to your response.
> >
> > Thanks

---

> > ### Author Response · Authors · 2024-11-24
> > **Please check our response**
> >
> > Dear reviewer ZGwT,
> >
> > The discussion period is ending very soon. We are yet to receive any feedback from you regarding our responses. We tried our best to address all your comments with additional experimental results. This is our sincere request to you to kindly check our responses and consider reassessing our paper.
> >
> > We look forward to your response.
> >
> > Thanks

---

> > ### Comment · Reviewer_ZGwT · 2024-11-26
> >
> > Thank you for the detailed response and experimental results. They addressed my concerns about the weaknesses. I will keep my score at 6.

---

> ### Author Response · Authors · 2024-11-25
> **Sincere request to check our responses at least once before the discussion ends tomorrow**
>
> Dear Reviewer ZGwT,
>
> We have been trying to reach out to you with a request to check our responses to your comments. We have meticulously addressed all your concerns with new sets of experiments. We are sure our responses will address your comments.
> Since the discussion period will end tomorrow, it is our sincere request to you to check our responses at least once and reassess our paper.
>
> Sincerely look forward to hearing from you.
>
> Thanks

---

### Author Response · Authors · 2024-11-20
**Confirmation of Rebuttal Submission**

Dear Reviewers,

We appreciate your effort to review our paper meticulously and provide us with your valuable suggestions. We have addressed all your comments and provided the additional results you suggested. We have also updated the paper (changes are highlighted in blue color) to accommodate these changes. We request you to kindly go through our responses and let us resolve any other query that you may have.

Thanks,

---

### Author Response · Authors · 2024-12-03
**Official Comments by Authors**

We sincerely thank all the reviewers for their valuable feedback during the discussion period and their comprehensive review of our work. We appreciate their recognition of the contributions of our work,

## The first calibration-free structured pruning technique

Our proposed method, PruneNet, is the *first* structured model pruning technique that does not need any calibration dataset for compression, wherein a data-independent importance scorer is used to prune the layers of an LLM. **All reviewers highlighted the importance of such calibration-free pruning methods**, particularly in data-private and low-resource settings, and a thorough discussion of the motivation behind such methods (**particularly with reviewer bk7b**) reaffirms the key contribution of our work. Completely removing calibration datasets also makes PruneNet a highly reusable and efficient pruning technique, as it bypasses the need to run expensive forward/backward passes of LLMs through the calibration data.

## Efficiency and transferability of the proposed method

With PruneNet, we can significantly reduce the model compression time by 50%. This efficiency, as highlighted in the paper, is a key advantage of PruneNet. It can compress an LLM at different ratios at once without retraining the policy learner, a feature that is particularly useful in practical applications. This adaptability to different edge device resources **has been acknowledged by both reviewers bk7b and ooUi**, further enhancing the method's practicality.

## Extensive comparison with baselines on a wide range of models and compression ratios

We have presented an extensive comparison of PruneNet with other competitive model pruning techniques, notably SliceGPT, in a variety of set-ups. This comparison, which includes scenarios with/without recovery fine-tuning (RFT) and using different calibration datasets (WikiText2, Alpaca), provides a comprehensive view of PruneNet's performance. We also compared the performance for a variety of model architectures, with the total number of parameters ranging from 125M to 13B. Such an extensive comparison showcases the robustness of our proposed method, even when baselines are calibrated with instruction-based calibration datasets such as Alpaca.

**All the reviewers (ZGwT, bk7b, ooUI, onBH) have acknowledged the results and thoroughness of our experiments**, providing positive feedback that reaffirms the quality and significance of our work.

## Robustness of PruneNet

We've also presented a detailed study of various components of the design choices in PruneNet. This includes the choice
of reward functions used in the policy learning process, the usage of FFN1 vs FFN2 layers for sampling, a comparison of a policy-based
pruner vs a static pruning method such as random sampling of indices and a comparison of stochastic and deterministic policies to be used for pruning. The results of these experiments motivate the design choices we made and highlight the robustness of PruneNet in various setups.

## Clarity of presentation

**Reviewer onBH commended the clarity of presentation in our paper.**

We've also thoroughly responded to every follow-up question for each reviewer. Here is a summary of all the responses:

* In response to reviewer ZGwT's follow-up questions, we presented an ablation study comparing PruneNet's performance against a simple random sampling approach (with PruneNet performing better). We highlighted the importance of sampling in our overall design (against simply choosing the top-k most important rows of a matrix).

* For reviewer bk7b, we presented a comparison of using different calibration datasets (WikiText2, Alpaca) with SliceGPT against PruneNet. While SliceGPT's performance is marginally better than PruneNet when it uses the Alpaca dataset for the LLaMa-2-7B model, the trend was the opposite for the Phi-2 model, exposing the inherent limitation of calibration-based pruning techniques (namely their sensitivity to the choice of calibration data).

* For reviewer ooUI, we presented the results of PruneNet with recovery fine-tuning. We conducted additional ablation experiments to highlight the importance of each design choice in PruneNet.

* For reviewer onBH, we presented a few more ablation results (FFN1 vs FFN2 layers, RFT), results of an implementation of PruneNet for matrices in the attention layer,  and the results of PruneNet for compression ratios larger than 40%.

Based on our responses, we observed an overall positive change in the scores:

* Reviewer onBH improved the soundness score of our paper from 3 to 4.
* Reviewers ooUI and bk7b improved the overall rating of our paper from 5 to 6.
* While reviewer ZGwT acknowledged that all of their concerns were addressed in our rebuttal, they maintained their original rating of 6.

We hope that our work motivates further research in developing more efficient model compression techniques in low-resource settings and makes a good contribution to the field.

---

### Meta-Review · Area_Chair_tz1v · 2024-12-13

**Metareview:**

The paper introduces PruneNet, a novel structured pruning method for LLMs that eliminates the need for calibration datasets. By learning a stochastic policy to preserve the spectral structure of model weights, PruneNet achieves superior efficiency and robustness compared to methods like SliceGPT, retaining high performance even at significant compression ratios. The ability to transfer policies across compression ratios and compress models in minutes highlights its practical value for low-resource and privacy-sensitive settings.

The paper is well-written, with strong empirical validation and thorough comparisons, addressing reviewer concerns effectively. While minor limitations, such as limited tests on larger models, were noted, the authors’ detailed rebuttals and additional results strengthened the work. Overall, this paper provides a significant contribution to model compression, offering an efficient and scalable solution. I recommend its acceptance.

**Additional Comments On Reviewer Discussion:**

During the rebuttal, reviewers raised concerns about the limited analysis of design choices, the scope of comparisons with baselines like SliceGPT, and the theoretical justification of certain metrics (e.g., KS distance). The authors addressed these by providing additional ablations, broader comparisons (including RFT and larger models), and clarified theoretical motivations and results. They also extended experiments to alternative settings, such as attention layers and higher compression ratios.

---

### Decision · Program_Chairs · 2025-01-22

Accept (Poster)